# Cohesin-dependence of neuronal gene expression relates to chromatin loop length

Lesly Calderon[1,2†], Felix D Weiss[2†], Jonathan A Beagan[3†], Marta S Oliveira[1,2], Radina Georgieva[1,2], Yi-Fang Wang[1,2], Thomas S Carroll[2], Gopuraja Dharmalingam[1,2], Wanfeng Gong[3], Kyoko Tossell[2], Vincenzo de Paola[2], Chad Whilding[1,2], Mark A Ungless[1,2], Amanda G Fisher[1,2], Jennifer E Phillips-Cremins[3,4,5]*, Matthias Merkenschlager[2]*

[1]MRC London Institute of Medical Sciences, Imperial College London, London, United Kingdom; [2]Institute of Clinical Sciences, Faculty of Medicine, Imperial College, London, United Kingdom; [3]Department of Bioengineering, University of Pennsylvania, Philadelphia, United States; [4]Epigenetics Program, Perelman School of Medicine, University of Pennsylvania, Philadelphia, United States; [5]Department of Genetics, Perelman School of Medicine, University of Pennsylvania, Philadelphia, United States

*For correspondence:
jcremins@seas.upenn.edu
(JEP-C);
matthias.merkenschlager@lms.
mrc.ac.uk (MM)

†These authors contributed
equally to this work

Competing interest: The authors
declare that no competing
interests exist.

Reviewing Editor: Jeremy J
Day, University of Alabama at
Birmingham, United States

**Abstract** Cohesin and CTCF are major drivers of 3D genome organization, but their role in neurons is still emerging. Here, we show a prominent role for cohesin in the expression of genes that facilitate neuronal maturation and homeostasis. Unexpectedly, we observed two major classes of activity-regulated genes with distinct reliance on cohesin in mouse primary cortical neurons. Immediate early genes (IEGs) remained fully inducible by KCl and BDNF, and short-range enhancer-promoter contacts at the IEGs *Fos* formed robustly in the absence of cohesin. In contrast, cohesin was required for full expression of a subset of secondary response genes characterized by long-range chromatin contacts. Cohesin-dependence of constitutive neuronal genes with key functions in synaptic transmission and neurotransmitter signaling also scaled with chromatin loop length. Our data demonstrate that key genes required for the maturation and activation of primary cortical neurons depend on cohesin for their full expression, and that the degree to which these genes rely on cohesin scales with the genomic distance traversed by their chromatin contacts.

## Editor's evaluation

Neurons use activity-responsive gene programs to shape cell-specific identity and respond appropriately to environmental stimuli. By combining elegant protein degradation and cell-specific knockout approaches with transcriptional profiling and chromatin structure analysis, this manuscript delineates the contributions of cohesin (a key protein responsible for genome structure and organization), in developmental and activity-dependent gene expression programs as well as chromatin reorganization. These results demonstrate that cohesin is required for the full expression of key genes required for the maturation and activation of cortical excitatory neurons, and reveal a tight correlation between cohesin effects and the genomic distance of higher-order chromatin loops.

## Introduction

Mutations in cohesin and CTCF cause intellectual disability in humans (*Deardorff et al., 2018*; *Gregor et al., 2013*; *Rajarajan et al., 2016*), and defects in neuronal gene expression (*Kawauchi et al.,*

*2009*; *Fujita et al., 2017*; *van den Berg et al., 2017*; *van den Berg et al., 2017*; *McGill et al., 2018*; *Yamada et al., 2019*; *Weiss et al., 2021*), neuronal morphology (*Fujita et al., 2017*; *McGill et al., 2018*; *Sams et al., 2016*), long-term potentiation (*Sams et al., 2016*; *Kim et al., 2018a*), learning, and memory (*McGill et al., 2018*; *Yamada et al., 2019*; *Sams et al., 2016*; *Kim et al., 2018b*) in animal models. In addition to mediating canonical functions in the cell cycle (*Nasmyth and Haering, 2009*), cohesin cooperates with CTCF to facilitate the spatial organization of the genome in the nucleus. Cohesin traverses chromosomal DNA in an ATP-dependent manner through a mechanism known as loop extrusion. This process generates self-interacting domains that are delimited by chromatin boundaries marked by CTCF and defined by an increased probability of chromatin contacts (*Fudenberg et al., 2016*; *Rao et al., 2014*; *Rao et al., 2017*; *Nora et al., 2017*; *Schwarzer et al., 2017*). The resulting organization of the genome into domains and loops is thought to contribute to the regulation of gene expression by facilitating appropriate enhancer-promoter interactions (*Dekker and Mirny, 2016*; *Merkenschlager and Nora, 2016*; *Beagan and Phillips-Cremins, 2020*; *McCord et al., 2020*) and its disruption can cause human disease (*Lupiáñez et al., 2015*; *Spielmann et al., 2018*; *Sun et al., 2018*), including neurodevelopmental disorders (*Won et al., 2016*). Recent studies which induced global cohesin loss on acute timescales resulted in the altered expression of only a small number of genes in a human cell line in vitro (*Rao et al., 2017*), while later time points after cohesin depletion in vivo revealed more pervasive disruption in expression (*Schwarzer et al., 2017*). We recently reported that cohesin loss has modest effects on constitutive gene expression in uninduced macrophages, but severely disrupted the establishment of new gene expression programs upon the induction of a new macrophage state (*Cuartero et al., 2018*). These data support a model in which cohesin-mediated loop extrusion is more important for the establishment of new gene expression rather than maintenance of existing programs (*Cuartero et al., 2018*). However, the applicability of this model across other cell types remains unclear, and the extent to which deficits in cohesin function alter neuronal gene expression remains a critical underexplored question.

Activity-regulated neuronal genes (ARGs) are defined by transcriptional induction in response to neuronal activity, and are important for cellular morphology, the formation of synapses and circuits, and ultimately for learning and memory (*Gallo et al., 2018*; *Kim et al., 2010*; *Malik et al., 2014*; *Greer and Greenberg, 2008*; *Tyssowski et al., 2018*; *Yap and Greenberg, 2018*). ARG induction is accompanied by acetylation of H3K27, as well as the recruitment of RNA polymerase 2, cohesin, and other chromatin binding proteins at ARGs and their enhancers (*Greer and Greenberg, 2008*; *Malik et al., 2014*; *Yamada et al., 2019*; *Yap and Greenberg, 2018*; *Tyssowski et al., 2018*; *Schaukowitch et al., 2014*; *Beagan et al., 2020*). For a subset of ARGs, long-range contacts between enhancers and promoters increase upon stimulation of post-mitotic neurons (*Schaukowitch et al., 2014*; *Sams et al., 2016*; *Beagan et al., 2020*). Among neuronal ARGs, IEGs and, late response genes (LRGs) are known to be activated on different time scales according to distinct mechanisms (*Tyssowski et al., 2018*). Moreover, IEGs and LRGs differ significantly in their looping landscape, as IEGs form fewer, shorter enhancer-promoter contacts compared to the complex, long-range interactions formed by many LRGs (*Beagan et al., 2020*). Given the importance of cohesin and regulatory looping interactions for brain function, there is a strong imperative to understand the functional role of cohesin-dependent loops in the establishment and maintenance of developmentally regulated and activity-stimulated neuronal gene expression programs.

Here, we establish an experimental system to address the role of cohesin in 3D genome organization and gene expression in non-dividing, post-mitotic neurons, independently of essential cohesin functions in the cell cycle. We employ developmentally regulated expression of Cre recombinase under the control of the endogenous *Neurod6* locus (referred to here as *Nex*^Cre according to *Goebbels et al., 2006*; *Hirayama et al., 2012*) to inducibly deplete the cohesin subunit RAD21 specifically in immature post-mitotic mouse neurons in vivo. Cohesin depletion in immature post-mitotic mouse neurons disrupted CTCF-based chromatin loops, and reduced the expression of neuronal genes related to synaptic transmission, neuronal development, adhesion, connectivity, and signaling, resulting in impaired neuronal maturation. Neuronal ARGs were pervasively deregulated in cohesin-deficient neurons, consistent with a model in which the establishment of new gene expression programs in both macrophages and neurons is driven in part by cohesin-mediated enhancer-promoter contacts (*Rajarajan et al., 2016*; *Yamada et al., 2019*; *Sams et al., 2016*; *Schaukowitch et al., 2014*; *Cuartero et al., 2018*). The cohesin-dependence of ARGs was confirmed in an inducible system that

allows for proteolytic cohesin cleavage in primary neurons (*Weiss et al., 2021*), providing temporal control over cohesin levels on a time scale similar to the establishment of neuronal gene expression programs upon neural stimulation, thus allowing us to disentangle the role for cohesin-mediated chromatin contacts in the maintenance of existing transcriptional programs versus the establishment of new transcriptional programs in neural circuits. Surprisingly, despite pervasive deregulation at baseline, most IEGs and a subset of LRGs remained fully inducible by KCl and BDNF stimulation after genetic or proteolytic depletion of cohesin.

Cohesin-dependent and -independent ARGs were distinguished not by the binding of cohesin or CTCF to their promoters (*Schaukowitch et al., 2014*; *Sams et al., 2016*), but instead by their 3D connectivity. LRGs that depended on cohesin for full inducibility engaged in longer chromatin loops than IEGs, or LRGs that remained fully inducible in the absence of cohesin. Unlike ARGs, the majority of neuronal genes that mediate synaptic transmission and neurotransmitter signaling are constitutively expressed. Nevertheless, as with ARGs, the reliance of these key neuronal genes on cohesin scaled with chromatin loop length. Consistent with a model where short-range enhancer-promoter loops can form in the absence of cohesin, we find that the enhancer activity and short-range enhancer-promoter contacts at the IEG *Fos* remained robustly inducible in cohesin-depleted neurons. Finally, re-expression of RAD21 protein in cohesin-depleted neurons re-established lost chromatin loops and restored wild-type expression levels of disrupted LRGs, and of constitutive neuronal genes engaged in long-range loops. Together, our data support a model where key neuronal genes required for the maturation and activation of primary neurons require cohesin for their full expression. The degree to which neuronal genes rely on cohesin scales with the genomic distance traversed by their chromatin loops, including loops connecting promoters with activity-induced enhancers.

## Results
### Conditional deletion of cohesin in immature post-mitotic neurons

To explore the role of cohesin in post-mitotic neurons, we deleted the essential cohesin subunit RAD21 (*Rad21*[lox], *Seitan et al., 2011*) in immature cortical and hippocampal neurons using *Nex*[Cre] (*Goebbels et al., 2006*; *Hirayama et al., 2012*). Explant cultures of wild type and *Rad21*[lox/lox] *Nex*[Cre] E17.5/18.5 cortex contained >95% MAP2[+] neurons, with <1% GFAP[+] astrocytes or IBA1[+] microglia (*Figure 1a*). Immunofluorescence staining showed the loss of RAD21 protein specifically in GAD67[-] neurons (*Figure 1b and c*). This was expected, as *Nex*[Cre] is expressed by excitatory but not by inhibitory neurons (*Goebbels et al., 2006*; *Hirayama et al., 2012*). Consistent with the presence of ~80% of GAD67[-] and ~20% GAD67[+] neurons in the explant cultures, *Rad21* mRNA expression was reduced by 75–80% overall (*Figure 1d*, left). There was a corresponding reduction in RAD21 protein (*Figure 1d*, middle). To focus our analysis on cohesin-deficient neurons we combined *Nex*[Cre]-mediated deletion of *Rad21* with *Nex*[Cre]-dependent expression of an epitope-tagged ribosomal subunit (*Rpl22*-HA RiboTag; *Sanz et al., 2009*). We verified that *Nex*[Cre]-induced RPL22-HA expression was restricted to RAD21-depleted neurons (*Figure 1—figure supplement 1a*) and performed high throughput sequencing of *Rpl22*-HA RiboTag-associated mRNA (RiboTag RNA-seq, *Supplementary file 1*). Comparison with total RNA-seq (*Supplementary file 2*) showed that *Nex*[Cre] RiboTag RNA-seq captured excitatory neuron-specific transcripts, such as *Slc17a7* and *Camk2a*. Transcripts selectively expressed in astrocytes (*Gfap, Aqp4, Mlc1*), microglia (*Aif1*), and inhibitory neurons (*Gad1, Gad2, Slc32a1*) were depleted from *Nex*[Cre] RiboTag RNA-seq (*Figure 1—figure supplement 1b*). RiboTag RNA-seq enabled an accurate estimate of residual *Rad21* mRNA, which was <5% in *Rad21*[lox/lox] *Nex*[Cre] cortical neurons (*Figure 1d*, right). These data show near-complete loss of *Rad21* mRNA and undetectable levels of RAD21 protein in *Nex*[Cre]-expressing *Rad21*[lox/lox] neurons. Analysis of chromatin conformation by Chromosome-Conformation-Capture-Carbon-Copy (5 C) showed a substantial reduction in the strength of CTCF-based chromatin loops after genetic (*Rad21*[lox/lox] *Nex*[Cre], *Figure 1e*) and after proteolytic cohesin depletion (RAD21-TEV, *Figure 1—figure supplement 2*; *Weiss et al., 2021*). Of note, restoration of RAD21 expression rescued CTCF-based chromatin loop formation (*Figure 1—figure supplement 2*). These data show that cohesin is directly linked to the strength of CTCF-based chromatin loops in primary in post-mitotic neurons.

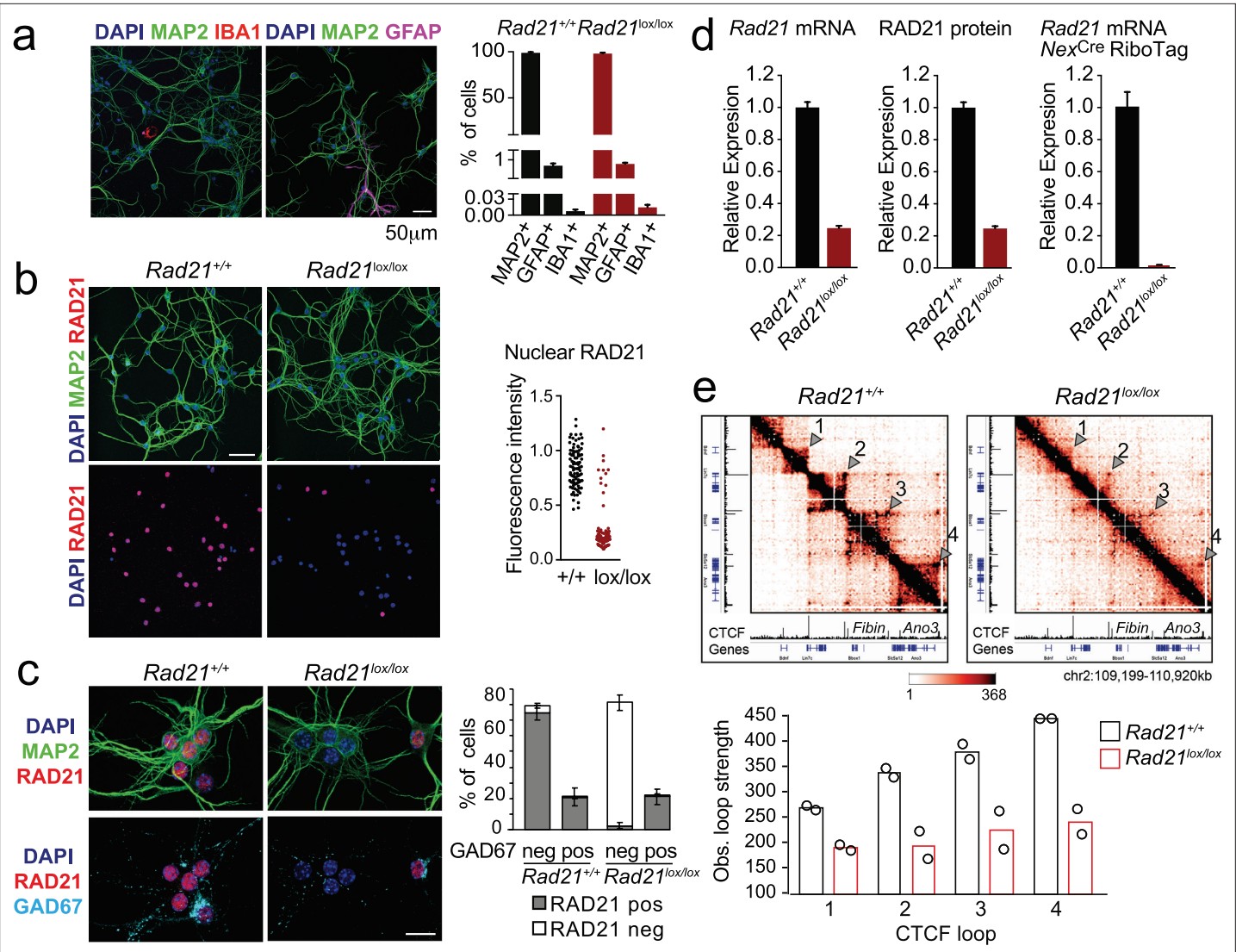

**Figure 1.** Conditional cohesin deletion in post-mitotic neurons. (**a**) E17.5–E18.5 cortices were dissociated and plated on poly-D-lysine. After 10 days, cultures were stained for pan neuronal (MAP2), astrocyte (GFAP), and microglia (IBA1) markers, and cell type composition was determined by quantitative analysis of immunofluorescence images. Based on 6 *Rad21*[+/+] *Nex*[Cre] and 8 *Rad21*[lox/lox] *Nex*[Cre] different samples analyzed in four independent experiments. (**b**) Immunofluorescence staining of *Rad21*[+/+] *Nex*[Cre] and *Rad21*[lox/lox] *Nex*[Cre] neuronal explant cultures for RAD21 and MAP2 (left) and distribution of RAD21 expression by MAP[+] neurons (right). Note the discontinuous distribution of RAD21 expression in *Rad21*[lox/lox] *Nex*[Cre] neurons. Three independent experiments per genotype. DAPI marks nuclei. Scale bar = 60 μm. (**c**) Immunofluorescence staining for RAD21, MAP2, and the marker of GABAergic inhibitory neurons, GAD67 (left). Distribution of RAD21 expression in GAD67[+] and GAD67[-] neurons (right). Note that the discontinuous distribution of RAD21 expression in *Rad21*[lox/lox] *Nex*[Cre] neuronal explant cultures is due to GAD67[+] GABAergic inhibitory neurons. Three independent experiments for *Rad21*[+/+] *Nex*[Cre] and six independent experiments for *Rad21*[lox/lox] *Nex*[Cre]. DAPI marks nuclei. Scale bar = 20 μm. (**d**) Quantitative RT-PCR analysis of *Rad21* mRNA expression in *Rad21*[+/+] *Nex*[Cre] and *Rad21*[lox/lox] *Nex*[Cre] cortical explant cultures (mean ± SEM, n=18). *Hprt* and *Ubc* were used for normalization (left). RAD21 protein expression in *Rad21*[+/+] *Nex*[Cre] and *Rad21*[lox/lox] *Nex*[Cre] cortical explant cultures was quantified by fluorescent immunoblots (mean ± SEM, n=6, a representative blot is shown in **Figure 1—figure supplement 1**) and normalized to LaminB (center). *Nex*[Cre] RiboTag RNA-seq of analysis of *Rad21* mRNA expression in *Rad21*[+/+] *Nex*[Cre] and *Rad21*[lox/lox] *Nex*[Cre] cortical explant cultures (right, three independent biological replicates). (**e**) 5C heat maps of *Rad21*[+/+] *Nex*[Cre] and *Rad21*[lox/lox] *Nex*[Cre] cortical explant cultures. Shown is a 1.72 Mb region covered by 5C analysis of chr2 107601077–110913077 (**Beagan et al., 2020**). One of two independent biological replicates with similar results. CTCF ChIP-seq in cortical neurons (**Bonev et al., 2017**) and mm9 coordinates are shown for reference. Arrowheads mark the position of CTCF-based loops. Results were consistent across two replicates and three chromosomal regions. Histograms below show the quantification of representative CTCF-based loops (arrowheads) in two independent biological replicates for control and *Rad21*[lox/lox] *Nex*[Cre] neurons.

The online version of this article includes the following source data and figure supplement(s) for figure 1:

**Source data 1.** *Figure 1*: Conditional cohesin deletion in post-mitotic neurons.

*Figure 1 continued on next page*

*Figure 1 continued*

**Figure supplement 1.** *Rad21* Nex<sup>Cre</sup> RiboTag validation.

**Figure supplement 2.** Restoration of cohesin rescues chromatin loops.

## Loss of cohesin from immature post-mitotic neurons perturbs neuronal gene expression

Using RiboTag RNA-seq to profile gene expression specifically in cohesin-depleted neurons we identified 1028 downregulated and 572 upregulated transcripts in *Rad21*<sup>lox/lox</sup> *Nex*<sup>Cre</sup> cortical neurons (*Figure 2a*, *Figure 2—figure supplement 1a*), with preferential deregulation of neuron-specific genes (p<2.2e-16, see methods). Gene ontology (*Figure 2b*) and gene set enrichment analysis (GSEA, *Figure 2—figure supplement 1b*) showed that downregulated genes in *Rad21*<sup>lox/lox</sup> *Nex*<sup>Cre</sup> neurons were enriched for synaptic transmission, neuronal development, adhesion, connectivity, and signaling (*Supplementary file 3*) and showed significant overlap with genes linked to human ASD (p=5.10E-15, *Figure 2—figure supplement 1c*; *Banerjee-Basu and Packer, 2010*). Upregulated genes showed no comparable functional enrichment (*Figure 2b*). Inducible ARGs such as *Fos*, *Arc*, and *Bdnf* are

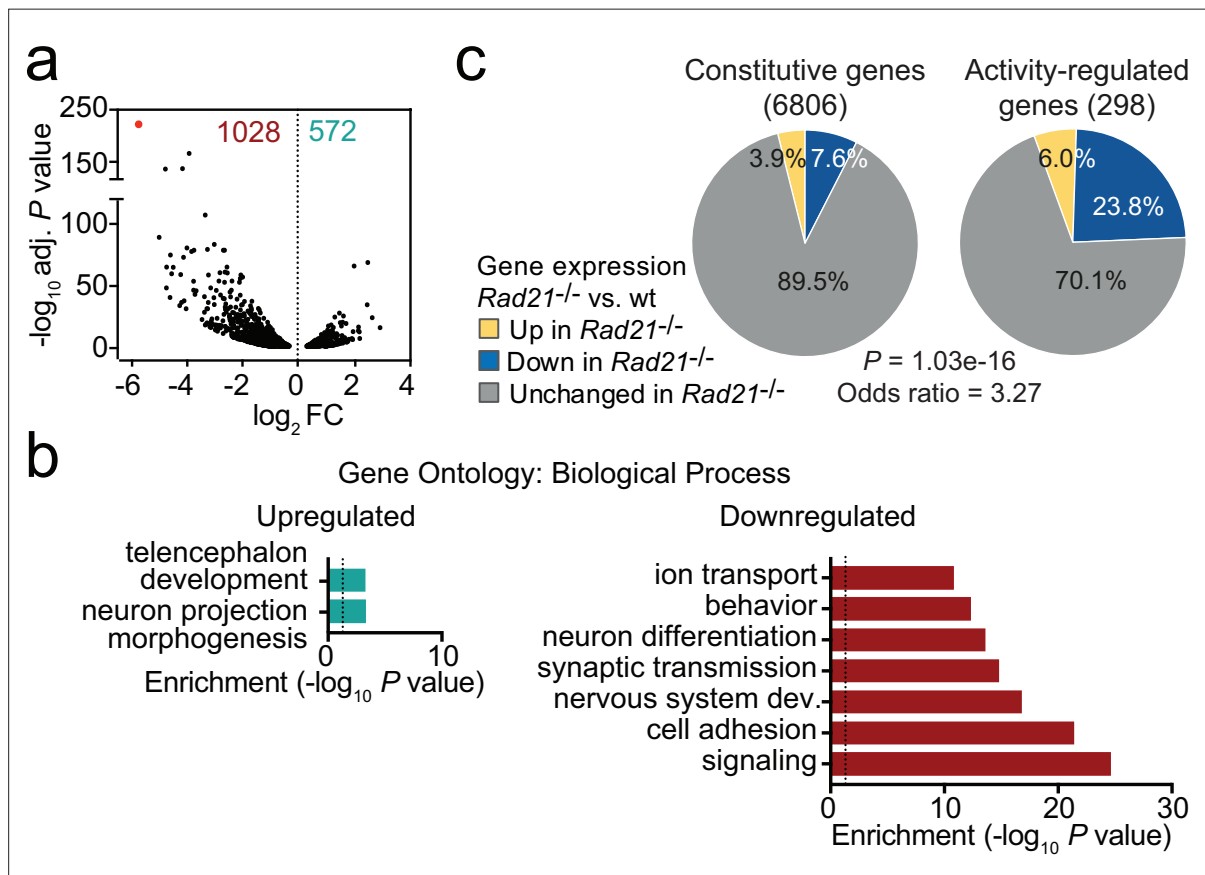

**Figure 2.** Loss of cohesin from immature post-mitotic neurons perturbs neuronal gene expression. (**a**) Volcano plot representing log2 fold-change (FC) versus significance (-log10 of adjusted p values) of downregulated genes (1028) and upregulated genes (572) in RiboTag RNA-seq of *Rad21*<sup>lox/lox</sup> *Nex*<sup>Cre</sup> versus *Rad21*<sup>+/+</sup> *Nex*<sup>Cre</sup> neurons (*Supplementary file 1*). Red marks *Rad21*. (**b**) Analysis of gene ontology of biological functions of deregulated genes in *Rad21*<sup>lox/lox</sup> *Nex*<sup>Cre</sup> neurons. Enrichment is calculated relative to expressed genes (*Supplementary file 3*). (**c**) The percentage of constitutive (adj. p>0.05 in KCl 1 hr versus TTX and KCl 6 hr versus TTX, see methods) and activity-regulated genes *Kim et al., 2010* found deregulated in *Rad21*<sup>lox/lox</sup> *Nex*<sup>Cre</sup> neurons in explant culture at baseline as determined by RiboTag RNA-seq. The p-value (Fisher Exact Test) and Odds ratio indicate that ARGs are more frequently deregulated than constitutive genes.

The online version of this article includes the following source data and figure supplement(s) for figure 2:

**Source data 1.** *Figure 2*: Loss of cohesin from immature post-mitotic neurons perturbs neuronal gene expression.

**Figure supplement 1.** Gene expression in *Rad21*<sup>lox/lox</sup> *Nex*<sup>Cre</sup> neurons.

activity-regulated by definition, and hence lowly expressed in basal conditions due to spontaneous synaptic activity. In *Rad21*[lox/lox]*Nex*[Cre] neurons, previously defined inducible ARGs (*Kim et al., 2010*) were more frequently downregulated than constitutively expressed genes (*Figure 2c*, p=1.03e-16, odds ratio = 3.27). These data show that immature post-mitotic neurons require cohesin to establish and/or maintain the correct level of expression of genes that support neuronal maturation, including the growth and guidance of axons, the development of dendrites and spines, and the assembly, function, and plasticity of synapses.

## A role for cohesin in the maturation of post-mitotic neurons

We next set out to examine the impact of cohesin deletion on the maturation of immature post-mitotic neurons. *Rad21*[lox/lox]*Nex*[Cre] embryos were found at the expected Mendelian ratios throughout gestation, however postnatal lethality was evident (*Figure 3—figure supplement 1a*). *Rad21*[lox/lox] *Nex*[Cre] cortical neurons did not show increased proliferation (*Figure 3—figure supplement 1b*), no upregulation of apoptosis markers or signs of DNA damage (*Figure 3—figure supplement 1b*) and no stress-related gene expression (*Figure 3—figure supplement 1c*). Brain weight (*Figure 3—figure supplement 1d*) and cellularity (*Figure 3—figure supplement 1e*) were comparable between *Rad21*[+/+] and *Rad21*[lox/lox] *Nex*[Cre] neocortex. The neuronal transcription factors TBR1, CTIP2, and CUX1 were expressed beyond the boundaries of their expected layers in *Rad21*[lox/lox] *Nex*[Cre] cortices, and deeper cortical layers appeared disorganized (*Figure 3a*). While the total numbers of TBR1[+] and CTIP2[+] neurons were comparable in wild type and *Rad21*[lox/lox] cortices, TBR1[+] and CTIP2[+] neurons were reduced in the subplate and increased in layers 6 and 7 *Rad21*[lox/lox] *Nex*[Cre] cortices (*Figure 3a*). These findings are consistent with the reported cohesin-dependence of neuronal guidance molecule expression (*Kawauchi et al., 2009*; *Remeseiro et al., 2012*; *Guo et al., 2015*) and migration (*van den Berg et al., 2017*). To assess the impact of cohesin on morphological maturation, we cultured cortical neurons in the presence of wild type glia (*Kaech and Banker, 2006*). Compared to freshly explanted E18.5 neurons (*Figure 3b*), neurons acquired considerable morphological complexity after 14 days in explant culture (*Figure 3c*). We sparsely labeled neurons with GFP to visualize processes of individual neurons (*Figure 3d*) and used Sholl analysis (*Sholl, 1953*) to quantitate the number of axonal crossings, the length of dendrites, the number of terminal points, the number of branch points and the number of spines in GAD67-negative *Rad21*[+/+] *Nex*[Cre] and GAD67-negative *Rad21*[lox/lox] *Nex*[Cre] neurons. Cohesin-deficient neurons displayed reduced morphological complexity across scales, with reduced numbers of axonal branch and terminal points (*Figure 3c and d*). In addition, *Rad21*[lox/lox] *Nex*[Cre] neurons showed reduced numbers of dendritic spines, the location of neuronal synapses (*Figure 3e*). Taken together, these data show that the changes in neuronal gene expression that accompany cohesin deficiency have a tangible impact on neuronal morphology, and that cohesin is required for neuronal maturation.

## Activity-regulated gene expression is sensitive to acute depletion of cohesin

Neuronal maturation and ARG expression are closely connected. The expression of inducible ARGs promotes neuronal maturation, morphological complexity, synapse formation, and connectivity. In turn, neuronal maturation, morphological complexity, synapse formation, and connectivity facilitate ARG expression (*Gallo et al., 2018*; *Kim et al., 2010*; *Malik et al., 2014*; *Greer and Greenberg, 2008*; *Tyssowski et al., 2018*; *Yap and Greenberg, 2018*). To address whether the downregulation of ARGs observed in *Rad21*[lox/lox] *Nex*[Cre] neurons was cause or consequence of impaired maturation we examined gene expression changes 24 hr after acute proteolytic degradation of RAD21-TEV depletion in post-mitotic neurons (*Weiss et al., 2021*). RNA-seq showed highly significant overlap in gene expression between acute degradation of RAD21-TEV and genetic cohesin depletion in *Rad21*[lox/lox] *Nex*[Cre] neurons (p<2.22e-16, odds ratio = 8.24 for all deregulated genes, odds ratio = 23.81 for downregulated genes; *Figure 4a*). Differentially expressed genes were enriched for ontologies related to synapse function, cell adhesion, and neuronal/nervous system development, and showed expression changes that were correlated across cohesin depletion paradigms (Synapse: p<2.22e−16, odds ratio = 11.26, $R_S$ = 0.62; Adhesion: p<2.22e−16, odds ratio = 15.21, $R_S$ = 0.6; Neuronal and nervous system development: p<2.22e−16, odds ratio = 9.92, $R_S$ = 0.58; *Figure 4b*). Notably, ARGs were enriched among deregulated genes in response to acute RAD21-TEV cleavage (p<2.22e-16, Odds Ratio =

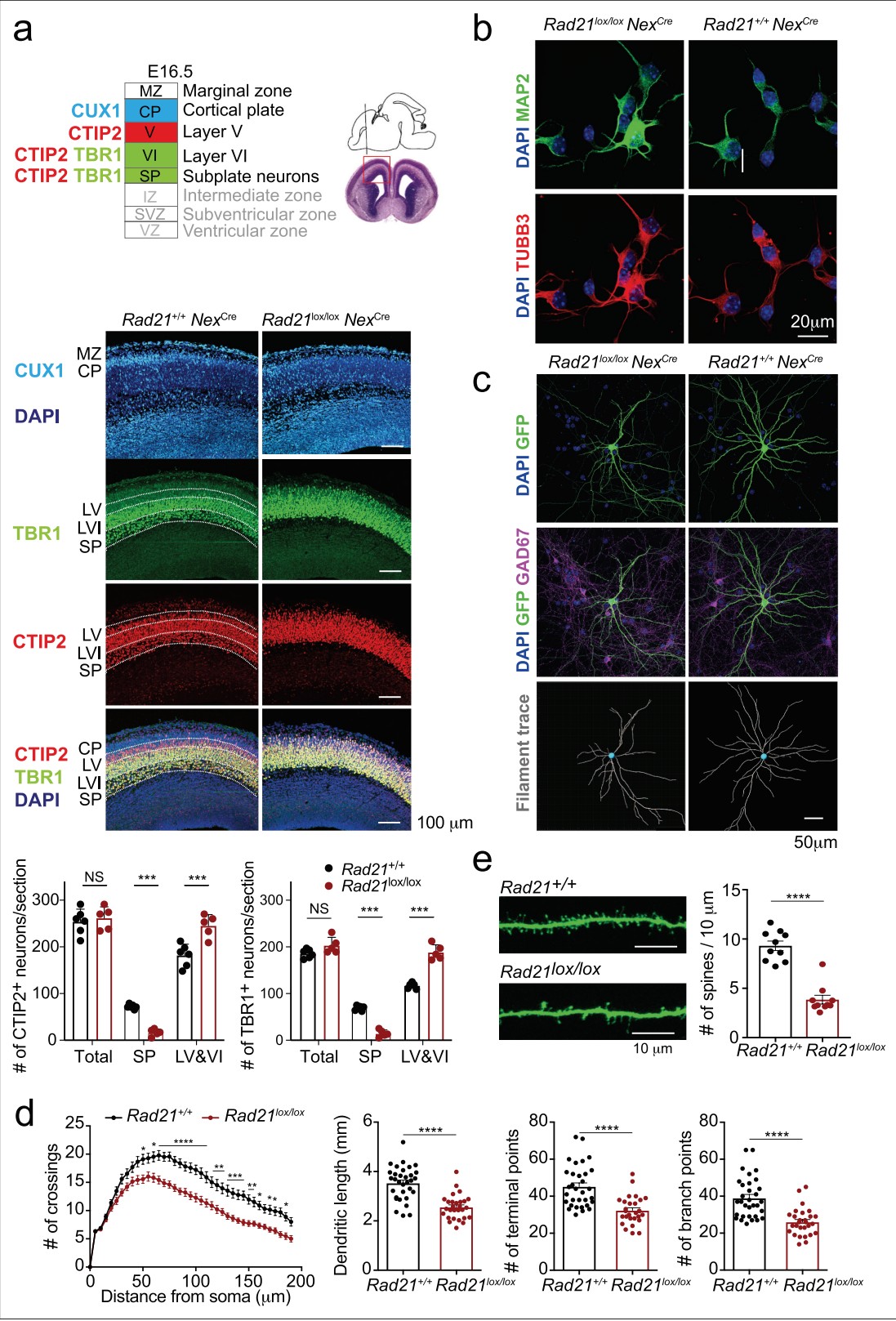

**Figure 3.** Cohesin contributes to the maturation of post-mitotic neurons. (**a**) Top: Schema of cortical layers (***Greig et al., 2013***) showing subplate (SP), layer 6 (VI), layer 5 (V), the cortical plate (CP), and the marginal zone (MZ). Middle: Immunofluorescence analysis of the neuronal transcription factors CUX1, TBR1, and CTIP2 at E16.5. Scale bar = 100 μm. Representative of three biological replicates. Bottom: Quantification of TBR1⁺ and CTIP2⁺ neurons in the subplate (SP) and in layers 5 and 6 (LV and VI). Neuron counts per 150 × 300 μm field are shown for five comparable sections from two embryos

*Figure 3 continued on next page*

*Figure 3 continued*

per genotype. Mean ± SE, *** adj. p<0.0001, two-way ANOVA with Sidak's multiple comparisons test. (**b**) Morphology of E18.5 neurons after 1d in explant culture. Immunofluorescence staining for the pan-neuronal marker MAP2, tubulin beta 3 (TUBB3), and DAPI. Scale bar = 20 μm. (**c**) Morphology of *Rad21*[+/+] *Nex*[Cre] and *Rad21*[lox/lox] *Nex*[Cre] cortical neurons in explant culture on rat glia (*Kaech and Banker, 2006*). Cultures were sparsely labeled with GFP to visualize individual cells and their processes, and stained for GAD67 to exclude GABAergic neurons. Dendritic traces of GFP[+] neurons. Scale bar = 50 μm. (**d**) Sholl analysis of *Rad21*[+/+] *Nex*[Cre] and *Rad21*[lox/lox] *Nex*[Cre] cortical neurons in explant cultures shown in (c). Shown is the number of crossings, dendritic length, terminal points, and branch points per 10 μm. Three independent experiments, 32 *Rad21*[lox/lox] *Nex*[Cre] and 28 *Rad21*[+/+] *Nex*[Cre] neurons. * adj. p<0.05, ** adj. p<0.01, *** adj. p<0.001, **** adj. p<0.0001. Scale bar = 10 μm. (**e**) Quantification of spines per 10 μm for *Rad21*[+/+] *Nex*[Cre] and *Rad21*[lox/lox] *Nex*[Cre] cortical neurons. Two independent experiments, 10 *Rad21*[lox/lox] *Nex*[Cre] and 10 *Rad21*[+/+] *Nex*[Cre] neuron. **** adj. p<0.0001. Scale bar = 10 μm.

The online version of this article includes the following source data and figure supplement(s) for figure 3:

**Source data 1.** *Figure 3*: Cohesin contributes to the maturation of post-mitotic neurons.

**Figure supplement 1.** Impact of cohesin loss in immature post-mitotic neurons in vivo.

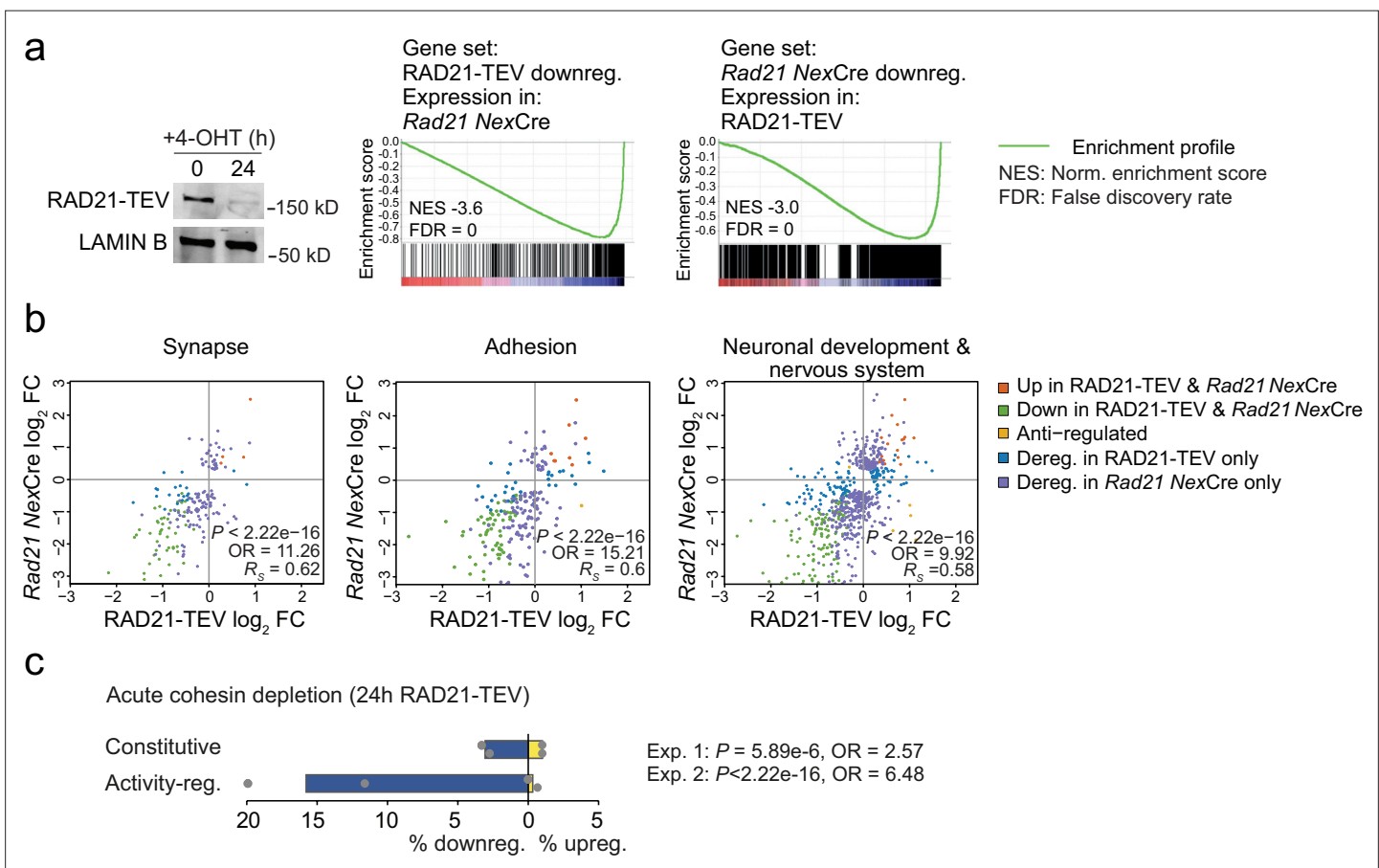

**Figure 4.** Acute cohesin depletion deregulates ARG expression. (**a**) Western blot documenting acute RAD21 depletion by 4-OHT-inducible RAD-TEV cleavage (left). GSEA of the gene set downregulated (DEseq2, adj. p<0.05) in RAD21-TEV neurons in *Rad21*[lox/lox] *Nex*[Cre] neurons (center). GSEA of genes downregulated in *Rad21*[lox/lox] *Nex*[Cre] neurons (DEseq2, adj. p<0.05) in RAD21-TEV neurons (right). NES: normalized enrichment score. FDR: false discovery rate. (**b**) Scatter plots of gene expression within aggregate GO terms, comparing RAD21-TEV with *Rad21*[lox/lox] *Nex*[Cre] neurons. Genes that were found deregulated in at least one of the genotypes are shown. p-values and odds ratios refer to the probability of finding the observed patterns of co-regulation by chance. $R_S$: Spearman's rank coefficient. (**c**) Deregulation of constitutive and activity-regulated genes 24 hr after acute cohesin depletion by inducible proteolytic cleavage of RAD21-TEV; adj. p<0.05 based on DEseq2 analysis of three RNA-seq replicates per experiment. Blue indicates downregulation and yellow indicates upregulation in RAD21-TEV versus wild type. Two independent experiments are shown (*Weiss et al., 2021* and *Supplementary file 4*).

6.48), and were preferentially downregulated (*Figure 4c*). Thus, we conclude that activity-regulated genes and genes that facilitate neuronal maturation and homeostasis are directly affected by the acute depletion of cohesin, and not just as a result of impaired neuronal maturation.

## Activity-regulated gene classes differ with respect to their reliance on cohesin

The disruption of baseline ARG expression suggested that cohesin-deficient neurons may be unable to induce the same activity-dependent gene expression program as wild-type neurons. To address this possibility, we performed RNA-seq of neuronal explant cultures treated either with tetrodotoxin +D-AP5 (TTX) alone to block neuronal signaling, or treated with TTX followed by 1 or 6 hr of sustained KCl exposure to induce neuronal depolarization. The fraction of constitutive genes deregulated in *Rad21*^lox/lox^ *Nex*^Cre^ versus control neurons remained similar across conditions (baseline, TTX, 1 hr and 6 hr KCl, *Figure 5a*, top). By contrast, approximately 50% (154/305) of ARGs (*Kim et al., 2010*) were downregulated in *Rad21*^lox/lox^ *Nex*^Cre^ versus control neurons under baseline conditions (*Figure 5—figure supplement 1a*). Of these, 76 were induced to control expression levels by KCl in *Rad21*^lox/lox^ *Nex*^Cre^ neurons, while 29% failed to reach control levels, and 16% were expressed at increased levels (*Figure 5—figure supplement 1a*). Multifactor analysis and hierarchical clustering (*Figure 5—figure supplement 1b*) showed that baseline ARG expression was more similar to TTX in *Rad21*^lox/lox^ *Nex*^Cre^ than in control neurons. This was confirmed by dendrogram distances (*Figure 5—figure supplement 1c*) and principal component analysis (*Figure 5—figure supplement 1d*), and statistically validated by the fraction of ARGs that changed expression between baseline and TTX conditions (49.5% in control neurons versus 28.5% in *Rad21*^lox/lox^ *Nex*^Cre^ neurons; p=5.89e-11, *Figure 5—figure supplement 1e*). Taken together, these data show that ARG expression is reduced in *Rad21*^lox/lox^ *Nex*^Cre^ neurons under baseline conditions, but remains responsive to activation. Among neuronal ARG classes, IEGs and LRGs are known to differ in their 3D connectivity in that LRGs engage in longer-range chromatin contacts than IEGs in primary cortical neurons (*Beagan et al., 2020*; see methods for the definition of IEGs and LRGs). However, the functional consequences of this difference in 3D connectivity remain to be explored. We therefore asked whether IEGs and LRGs differ with respect to their reliance on cohesin. While IEGs were induced to at least wild-type levels in cohesin-deficient neurons by stimulation with KCl (*Figure 5a*) or BDNF (*Figure 5b*), a substantial fraction of LRGs remained downregulated in cohesin-deficient neurons across conditions (*Figure 5c*). To test whether the expression of LRGs that remained downregulated in cohesin-deficient neurons across conditions was indeed cohesin-dependent, we restored RAD21 levels following transient proteolytic cleavage of RAD21-TEV. We found that the expression of LRGs was rescued by restoration of RAD21 (*Figure 5d*). These data show that there are two classes of ARGs: (i) IEGs/LRGs that exhibit altered baseline expression but can fully regain expression in response to activation, and (ii) a subset of LRGs that remain deregulated in response to activation.

## Cohesin-dependent neuronal genes have longer chromatin loops

To explore features that may explain why a subset of ARG requires cohesin for full expression we analyzed ChIP-seq data for cohesin and CTCF binding to ARG promoters in wild-type neurons. ARG promoters are enriched for binding of CTCF (OR = 1.621, p=0.022, two-tailed Fisher Exact Test), the cohesin subunit RAD21 (OR = 1.866, p=0.005), and cohesin in the absence of CTCF (cohesin-non-CTCF, OR = 1.621, p=0.022) compared to non-ARGs expressed in cortical neurons. Within the ARG gene set, however, there were no significant differences in CTCF, RAD21, or RAD21-non-CTCF ChIP-seq binding at the promoters of IEGs (which as a group remained fully inducible in cohesin-deficient neurons), the subset of LRGs that were downregulated across conditions in *Rad21* NexCre neurons (both TTX and 6 hr KCl, adj p<0.05), and LRGs that remained fully inducible across conditions in *Rad21* NexCre neurons (both TTX and 6 hr KCl, adj p>0.05, *Figure 6—figure supplement 1a*). Therefore, while ARG promoters are enriched for CTCF, RAD21, and cohesin-non-CTCF binding, this binding is not predictive of which ARGs remain fully inducible, and which are downregulated in cohesin-deficient neurons.

To test whether cohesin-dependence of ARG regulation might instead be linked to the genomic range of chromatin loops formed by these genes we analyzed high-resolution cortical neuron Hi-C data (*Bonev et al., 2017*). We found that the subset of LRGs that required cohesin for full expression

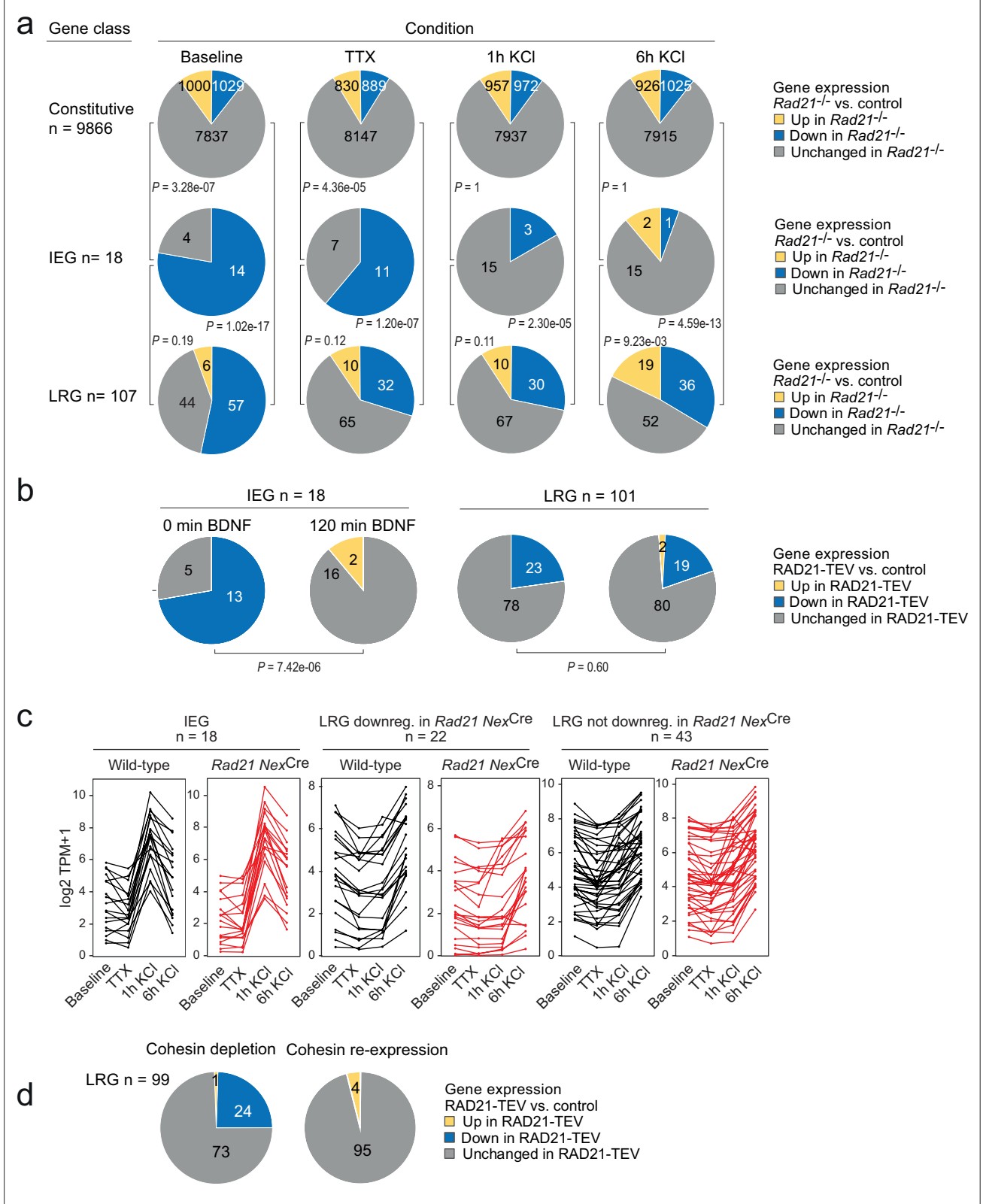

**Figure 5.** Activity-regulated neuronal gene (ARG) classes differ in their reliance on cohesin. (**a**) Pie charts show the expression of constitutive genes (top), immediate early genes (IEGs) (center), and late response genes (LRGs) (bottom) in *Rad21* NexCre neurons under four different conditions: baseline, TTX and D-AP5 (TTX), and in response to KCl stimulation for 1 hr or 6 hr. Numbers of expressed constitutive genes, IEGs, and LRGs are given on the left. Note that KCl stimulation normalizes the expression of most (10 out of 11) IEGs downregulated in TTX, but a fraction of LRGs remain downregulated.

*Figure 5 continued on next page*

*Figure 5 continued*

p-values test the prevalence of deregulated genes in each class under each condition, two-sided Fisher exact test. (**b**) Pie charts show the expression of IEGs (top) and LRGs (bottom) in RAD21-TEV neurons under baseline conditions 24 hr after ERt2-TEV induction and in response to BDNF (120 min). RAD21-TEV cleavage led to the downregulation (adj. p<0.05) of 13 out of 18 expressed IEGs and of 23 out of 101 expressed LRGs. p-values test the prevalence of downregulated IEGs and LRGs with and without BDNF stimulation, two-tailed Fisher exact test. Note that BDNF stimulation reversed the downregulation of IEGs but not LRGs in cohesin-depleted neurons. (**c**) Strip plots depict the expression of IEGs, LRGs that are downregulated in *Rad21* NexCre neurons compared to control across conditions (TTX and 6 hr KCl), and LRGs that are not downregulated in *Rad21* NexCre neurons compared to control across conditions. (**d**) Transient cohesin depletion and re-expression as in *Figure 1—figure supplement 2*. Pie charts show the expression of LRGs in RAD21-TEV relative to control neurons. 24 out of 97 LRGs expressed in RAD21-TEV neurons were downregulated 24 hr after Dox-dependent TEV induction (adj. p<0.05). The downregulation of LRGs was reversible upon Dox washout and restoration of RAD21 expression.

The online version of this article includes the following source data and figure supplement(s) for figure 5:

**Source data 1.** *Figure 5*: Activity-regulated neuronal gene (ARG) classes differ in their reliance on cohesin.

**Figure supplement 1.** Gene expression and genotype interaction analysis.

in TTX and full induction by 6 hr KCl (adj p<0.05) formed significantly longer Hi-C loops than IEGs and LRGs that were not deregulated across conditions (TTX and 6 hr KCl, adj p>0.05) in *Rad21* NexCre neurons (*Figure 6a*). Chromatin loops with CTCF binding at one or both loop anchors were also significantly longer for cohesin-dependent LRGs than for IEGs and cohesin-independent LRGs (*Figure 6a*, red). Chromatin loops that connect promoters with inducible enhancers also tended to span larger genomic distances at downregulated LRGs compared to IEGs or non-deregulated LRGs, even though due to the limited numbers of enhancer-promoter loops associated with each LRG class, these trends do not reach statistical significance (*Figure 6a*, blue). Overall, the degree to which ARGs depend on cohesin for their correct expression correlates with the length of chromatin loops they form.

We next addressed whether cohesin/CTCF binding or chromatin loop length were important factors for the impact of cohesin on the expression of additional neuronal genes. We focused on the neuronal GO terms 'synaptic transmission' (GO:0007268) and 'glutamate receptor signaling pathway' (GO:0007215) because these gene ontologies were highly enriched among downregulated genes both in acute RAD21-TEV cleavage and genetic cohesin depletion, and remained enriched across conditions in *Rad21* NexCre neurons (GO:0007268: 87 of 519 genes downregulated at 6 hr KCl, adj. p<0.05, p-value for enrichment = 2.99E-07; GO:0007215: 21 of 68 genes downregulated at 6 hr KCl, adj p<0.05, p-value for enrichment = 6.46E-06). The majority of synaptic transmission and glutamate receptor signaling genes are classified as constitutive (62.7% of expressed and 59.8% of downregulated GO:0007268 and GO:0007215 genes across conditions), rather than activity-regulated (1.8% of expressed and 1.2% of downregulated GO:0007268 and GO:0007215 genes cross conditions), thus complementing the analysis of ARGs. Of note, restoration of RAD21 after transient depletion in the RAD21-TEV system rescued the expression of 90% (70 of 77) downregulated synaptic transmission and glutamate receptor signaling genes (*Figure 6b*), confirming that their expression was indeed dependent on cohesin.

As described above for ARGs, the TSSs of neuronal GO term genes related to synaptic transmission and glutamate receptor signaling were enriched for binding of CTCF (OR = 1.409, p<0.0001, two-tailed Fisher Exact Test), RAD21 (OR = 1.745, p<0.0001), and cohesin-non-CTCF (OR = 1.585, p<0.0001) cCompared to the remaining expressed genes in cortical neurons. However, and again as described above for ARGs, the binding of CTCF, RAD21, or cohesin-non-CTCF to the promoters of neuronal GO term genes did not predict which neuronal GO term genes remained expressed at wild-type levels, and which were downregulated in *Rad21* NexCre neurons (*Figure 6—figure supplement 1b*).

Notably, however, synaptic transmission and glutamate receptor signaling genes that were downregulated in *Rad21*$^{lox/lox}$ *Nex*$^{Cre}$ neurons engaged in significantly longer-range chromatin loops than genes that were either not deregulated or upregulated (*Figure 6c*, black, strip plots show the expression of the depicted GO term genes in control and *Rad21*$^{lox/lox}$ *Nex*$^{Cre}$ neurons). As with ARGs, this pattern extended to Hi-C loops with CTCF binding at one or both loop anchors (*Figure 6c*, red). The subset of synaptic transmission and glutamate receptor signaling genes that were downregulated in *Rad21*$^{lox/lox}$ *Nex*$^{Cre}$ neurons formed significantly longer Hi-C loops connecting promoters and enhancers than genes in the same gene ontologies that were not deregulated (p<0.0001) or upregulated (p<0.0001) in *Rad21* NexCre neurons (*Figure 6c*, blue). These data extend the relationship

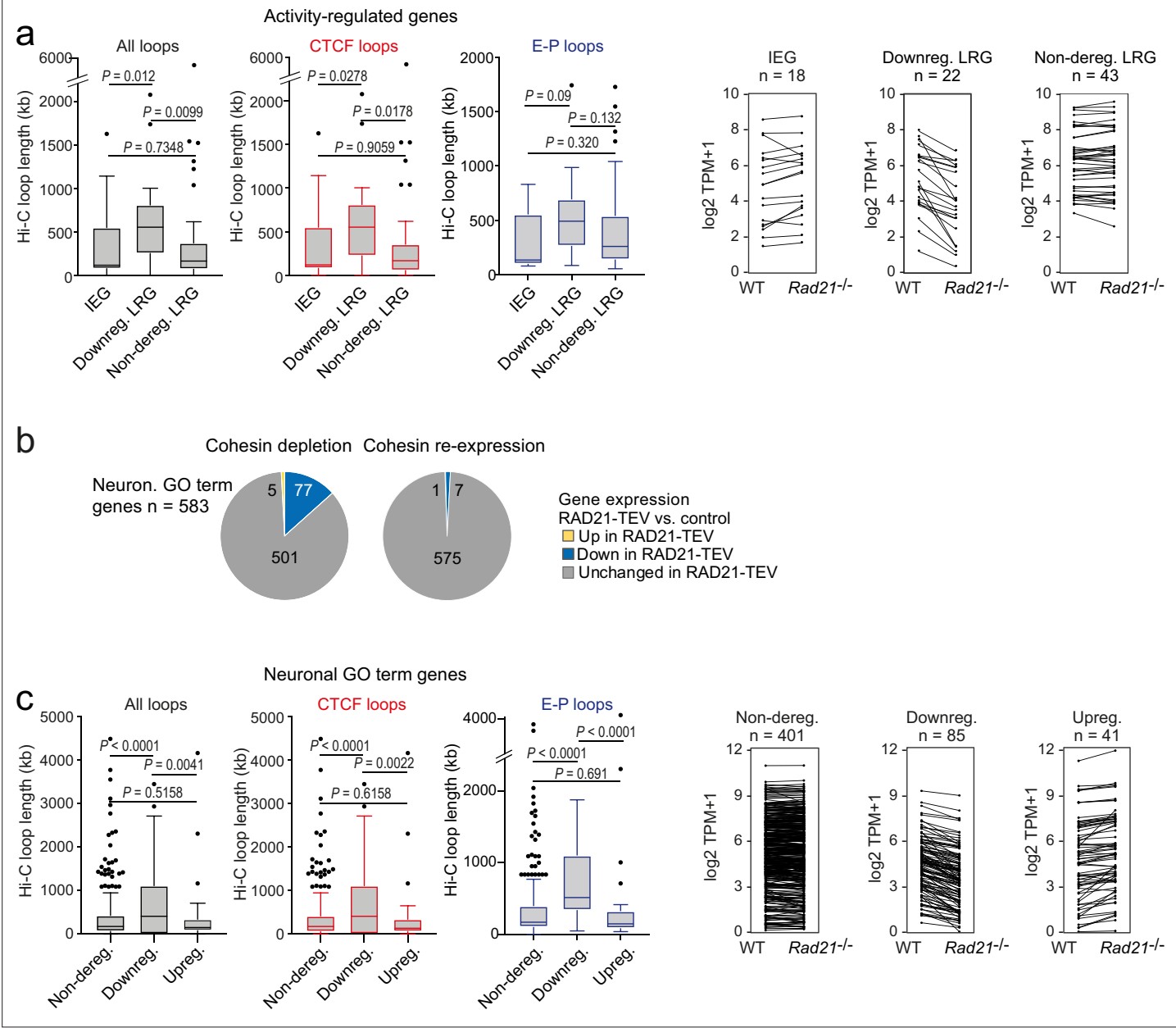

**Figure 6.** The genomic distance traversed by chromatin contacts formed by neuronal genes predicts whether or not cohesin is required for their full expression. (**a**) The span of Hi-C loops (left), Hi-C loops with CTCF bound to at least one of the loop anchors (middle) and Hi-C loops between promoters and inducible enhancers (right) for Immediate early genes (IEGs) (n=18) and late response genes (LRGs) downregulated in *Rad21 Nex*Cre versus control neurons in both resting (TTX) and activation conditions (6 hr KCl, adj p<0.05 in both TTX and 6 hr KCl conditions, n=22, 'Downreg. LRG'), and LRGs not deregulated in either resting (TTX) or activation conditions (6 hr KCl, adj p>0.05 in both TTX and 6 hr KCl conditions) in *Rad21 Nex*Cre relative to control neurons (adj. p>0.05, n=43, 'Non-dereg. LRG'). Box plots show the longest loop for each gene rather than average loop length, as Hi-C loop calling at 10 kb resolution precludes detection of loops <40 kb (***Beagan et al., 2020***). However, analysis of average loop length confirmed that downregulated genes form longer loops than non-deregulated genes among both ARGs and neuronal GO term genes (p=0.0056 and p<0.0001, respectively). Genes without loops are included except for analysis of enhancer loops. Box plots show the longest loop recorded for each gene. Boxes show upper and lower quartiles and whiskers show 1.5 of the interquartile range. p-values were determined by non-parametric Kolmogorov-Smirnov test. Strip plots depict the expression of IEGs, downregulated LRGs, and non-deregulated LRGs in wild-type and *Rad21* NexCre neurons. (**b**) Transient cohesin depletion and re-expression as in ***Figure 1—figure supplement 2***. Pie charts show the expression of neuronal genes related to synaptic transmission (GO:0007268) and glutamate receptor signaling pathway (GO:0007215). 77 out of 583 expressed genes in these GO terms (Neur. GO term genes) were downregulated 24 hr after Dox-dependent TEV induction (adj. p<0.05). The downregulation of 76 of 77 neuronal GO term genes was reversible upon Dox washout and restoration of RAD21 expression, an additional 6 genes were downregulated after Dox washout but not at 24 hr of TEV induction. (**c**) The span of Hi-C loops (left), Hi-C loops with CTCF bound to at least one of the loop anchors (middle) and Hi-C loops between

*Figure 6 continued*

promoters and constitutive or inducible enhancers (right) for genes in the neuronal GO terms synaptic transmission and glutamate receptor signaling. Gene expression in *Rad21 Nex*Cre versus control neurons was assessed in both resting and activation conditions: not deregulated in TTX or 6 hr KCl, n=401, Downregulated in both TTX and 6 hr KCl, n=85, Upregulated in both TTX and 6 hr KCl, n=41. Genes do not form loops are included except for analysis of enhancer loops. Boxes show upper and lower quartiles and whiskers show 1.5 of the interquartile range. p-values were determined by non-parametric Kolmogorov-Smirnov test. Strip plots show the expression of the depicted GO term genes in control and *Rad21* NexCre neurons.

The online version of this article includes the following source data and figure supplement(s) for figure 6:

**Source data 1.** *Figure 6*: The genomic distance traversed by chromatin contacts formed by neuronal genes predicts whether or not cohesin is required for their full expression.

**Figure supplement 1.** Promoter binding of CTCF or cohesin does not distinguish cohesin-dependent from cohesin-independent genes.

between chromatin loop length and cohesin-dependent expression from ARGs to constitutively expressed neuronal genes.

## Cohesin is not essential for short-range loops between inducible enhancers and promoters at the activity-regulated *Fos* and *Arc* loci

Typical IEG enhancers such as *Fos* and *Arc* enhancers are fully accessible prior to stimulation (*Carullo et al., 2020*), and show increase H3K27ac and eRNA transcription in response to neuronal activation (*Malik et al., 2014*; *Kim et al., 2010*; *Beagan et al., 2020*; *Joo et al., 2016*; *Carullo et al., 2020*). To examine the contribution of enhancer activation and enhancer-promoter contacts to inducible ARG expression in *Rad21*$^{lox/lox}$ *Nex*$^{Cre}$ neurons we focused on the immediate early response gene *Fos*. *Fos* expression in *Rad21*$^{lox/lox}$ *Nex*$^{Cre}$ neurons was reduced at baseline and in the presence of TTX, but *Fos* expression remained fully inducible when *Rad21*$^{lox/lox}$ *Nex*$^{Cre}$ neurons were stimulated with KCl (*Figure 7a*) or with BDNF (*Figure 7—figure supplement 1*). The transcription of neuronal genes is controlled by neuronal enhancers (*Rajarajan et al., 2016*; *Yamada et al., 2019*; *Sams et al., 2016*; *Kim et al., 2010*; *Malik et al., 2014*; *Schaukowitch et al., 2014*; *Beagan et al., 2020*). Neuronal *Fos* enhancers in particular have been extensively characterized (*Joo et al., 2016*; *Beagan et al., 2020*), and interference with *Fos* enhancers precludes full induction of *Fos* gene expression (*Joo et al., 2016*). *Fos* enhancers 1, 2, and 5 are known to undergo activation-induced acetylation of H3K27 (H3K27ac) and active eRNA transcription in response to KCl stimulation (*Joo et al., 2016*). We found that activation-induced transcription of *Fos* enhancers did remain intact in *Rad21*$^{lox/lox}$ *Nex*$^{Cre}$ neurons (*Figure 7b*) and activation-induced H3K27ac of *Fos* enhancers was also preserved (*Figure 7c*).

Given that *Fos* can be fully induced, and *Fos* enhancers are activated in the absence of cohesin, we next set out to understand if *Fos* can form previously reported looping interactions with its activity-stimulated enhancers (*Beagan et al., 2020*). We conducted 5 C to generate 10 kilobase resolution maps of chromatin loops around key IEGs and LRGs. Consistent with previous data (*Beagan et al., 2020*), *Fos* enhancers 1 and 2, which are located ~18 and ~ 38.5 kb upstream of the *Fos* TSS, showed inducible chromatin contacts with the *Fos* promoter that formed rapidly in response to activation of wild-type neurons (*Figure 7d*). Unexpectedly, *Rad21*$^{lox/lox}$ *Nex*$^{Cre}$ neurons retained the ability to robustly and dynamically induce loops between the *Fos* promoter and *Fos* enhancers 1 and 2 (*Figure 7d*). Quantification showed that inducible enhancer-promoter contacts at the *Fos* locus were of comparable strength in control and *Rad21*$^{lox/lox}$ *Nex*$^{Cre}$ neurons (*Figure 7e*, top and center), while a structural CTCF-based loop surrounding the *Fos* locus was substantially weakened (*Figure 7e*, bottom). Together these results reveal the surprising finding that the critical activity-stimulated IEG *Fos* can fully activate expression levels and form robust enhancer-promoter loops in the absence of cohesin.

We also examined the long-range regulatory landscape of the IEG *Arc*. The expression of the IEG *Arc* was reduced in *Rad21*$^{lox/lox}$ *Nex*$^{Cre}$ neurons at baseline, but, like *Fos*, *Arc* remained inducible by stimulation with KCl (*Figure 7—figure supplement 2a*, top) and BDNF (*Figure 7—figure supplement 2a*, bottom). Stimulation of wild-type neurons is known to trigger the formation of chromatin contacts between the *Arc* promoter and an *Arc*-associated activity-induced enhancer located ~15 kb downstream of the *Arc* TSS (*Beagan et al., 2020*; *Figure 7—figure supplement 2b*). As observed for *Fos*, *Arc* promoter-enhancer contacts were retained in *Rad21*$^{lox/lox}$ *Nex*$^{Cre}$ neurons (*Figure 7—figure supplement 2b, c*), while CTCF-based loops surrounding the *Arc* locus were weakened (*Figure 7—figure supplement 2b, c*). In contrast to control neurons, *Arc* promoter-enhancer contacts became at

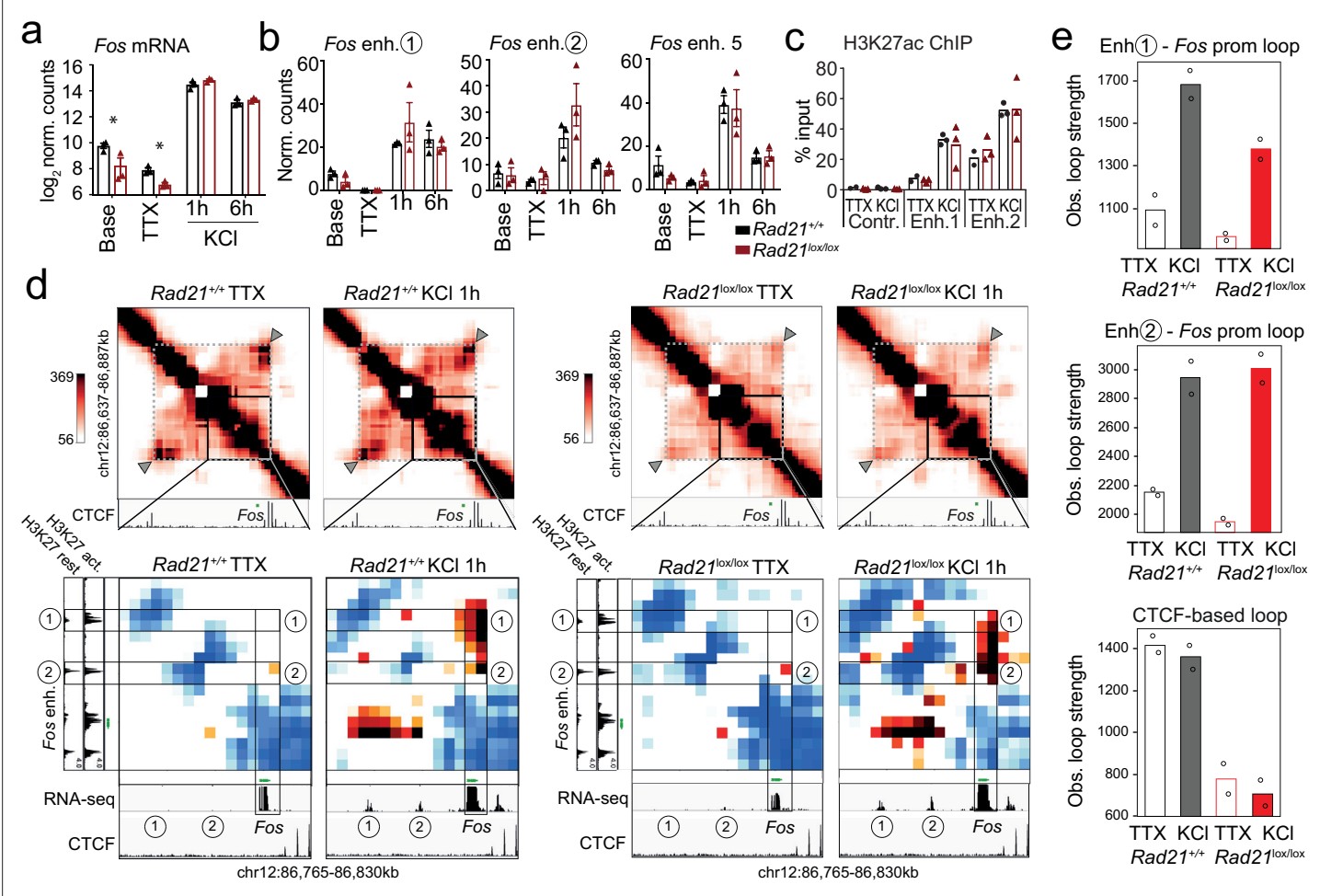

**Figure 7.** *Fos* enhancer-promoter contacts are robustly induced in cohesin-deficient neurons. (**a**) Expression of the immediate early genes (IEG) *Fos* at baseline, after TTX/D-AP5 (TTX), and KCl-stimulation (left, mean log2-transformed counts from 3 biological replicates, * adj. p<0.05). (**b**) Enhancer transcripts in control and *Rad21*^lox/lox *Nex*^Cre neurons were quantified based on normalized RNA-seq reads within 1 kb of the enhancer. An intergenic region on chr11 was used as a negative control (71.177.622–71.177.792) . (**c**) H3K27ac ChIP normalized to H3 in control and *Rad21*^lox/lox *Nex*^Cre neurons at a control site, *Fos* enhancer 1 and *Fos* enhancer 2 after TTX/D-AP5 (TTX) or 1 hr KCl (KCl). (**d**) Interaction frequency (top) and interaction score (bottom) heatmaps of the region immediately surrounding *Fos* obtained by 5C analysis of chr12 86201802–87697802 (**Beagan et al., 2020**). Black frames highlight interactions between the *Fos* gene and upstream enhancers 1 and 2. CTCF ChIP-seq in cortical neurons (**Bonev et al., 2017**) is shown for orientation and H3K27ac ChIP-seq in inactive (TTX-treated) and activated neurons is shown to annotate enhancer regions (**Beagan and Phillips-Cremins, 2020**; **Beagan et al., 2020**). RNA-seq in TTX-treated and 1 hr KCl-activated control and *Rad21*^lox/lox *Nex*^Cre neurons shows KCl-inducible transcription of *Fos* enhancers in wild -type and cohesin-deficient neurons. Two independent biological replicates are shown in **Figure 7— figure supplement 3a**. (**e**) Quantification of the interaction frequencies between the *Fos* promoter and *Fos* enhancer 1 (top), the *Fos* promoter and *Fos* enhancer 2 (middle), and CTCF-marked boundaries of the sub-TAD containing *Fos* (bottom, grey arrowhead). Two replicates per genotype and condition.

The online version of this article includes the following source data and figure supplement(s) for figure 7:

**Source data 1.** *Figure 7*: Fos enhancer-promoter contacts are robustly induced in cohesin-deficient neurons.

**Figure supplement 1.** Inducible gene expression in cohesin-deficient neurons.

**Figure supplement 2.** Contacts between the *Arc* promoter and an inducible enhancer in wild -type and cohesin-deficient neurons.

**Figure supplement 3.** Replicate 5 C experiments.

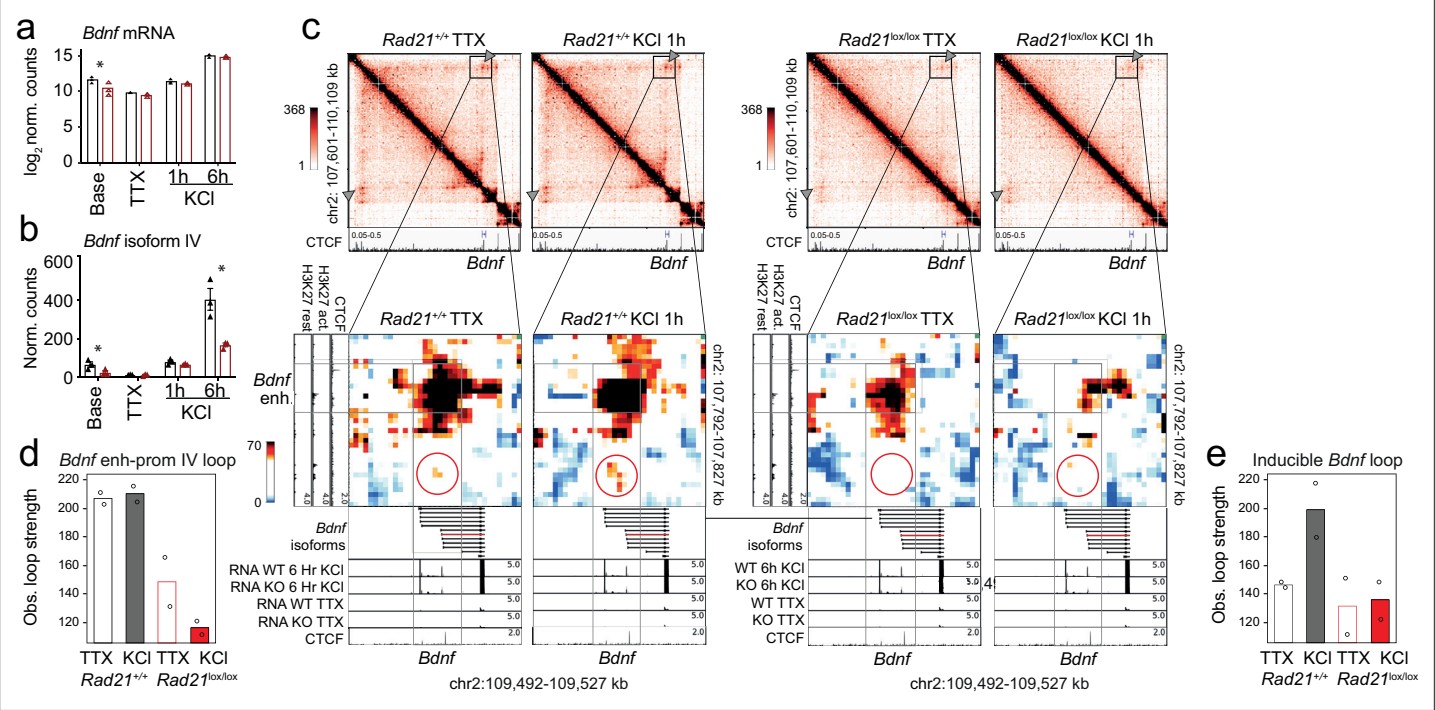

**Figure 8.** *Bdnf* enhancer-promoter contacts are weakened in the absence of cohesin. (**a**) Total *Bdnf* transcripts at baseline, after TTX/D-AP5 (TTX), and KCl-stimulation (left, mean log2-transformed counts from three biological replicates, * adj. p<0.05). (**b**) *Bdnf* promoter IV transcripts at baseline, after TTX/D-AP5 (TTX), and KCl-stimulation (left, mean log2-transformed counts from three biological replicates, * adj. p<0.05). (**c**) Interaction frequency (top) and interaction score (bottom) heatmaps of the *Bdnf* region obtained by 5C analysis of chr2 107601077-110913077 (*Beagan et al., 2020*). CTCF ChIP-seq in cortical neurons (*Bonev et al., 2017*) and the position of *Bdnf* are displayed (top). Below: Zoom-in of constitutive *Bdnf* enhancer-promoter loop (gray frame). Shown on the side is H3K27ac ChIP-seq in resting and activated neurons, marking an activity-dependent enhancer, and CTCF ChIP-seq. RNA-seq in resting and activated wild-type and cohesin deficient neurons and CTCF ChIP-seq are shown underneath. A circle marks an inducible 1.68Mb 5C loop between *Bdnf* and an activity-induced enhancer (*Bdnf* enhancer 1 in *Beagan et al., 2020*). (**d**) Quantification of 5C interaction frequencies between *Bdnf* promoter IV and the activity-dependent enhancer. (**e**) Quantification of inducible 5C loop between *Bdnf* and *Bdnf* enhancer 1 (*Beagan et al., 2020*).

The online version of this article includes the following figure supplement(s) for figure 8:

**Figure supplement 1.** Quantification of observed and distance-corrected loop strength.

least partially independent of activation in *Rad21*^lox/lox *Nex*^Cre neurons (***Figure 7—figure supplement 2b, c***). Hence, cohesin is required for the correct baseline expression of ARGs, but largely dispensable for inducible transcription and for specific enhancer-promoter contacts at the IEGs *Fos* and *Arc*.

## Cohesin-dependent long-range looping at the *Bdnf* locus

In contrast to IEGs *Fos* and *Arc*, the LRG *Bdnf* has at least eight promoters that initiate transcription of distinct mRNA transcripts, all of which contain the entire open reading frame for the BDNF protein (***Aid et al., 2007***). *Bdnf* promoter IV is specifically required for the neuronal activity-dependent component of *Bdnf* transcription in mouse cortical neurons (***Hong et al., 2008***). While overall *Bdnf* transcript levels were significantly reduced in *Rad21*^lox/lox *Nex*^Cre neurons only at baseline (log 2 FC = –1.16 adj p=0.003, ***Figure 8***), *Bdnf* transcripts from the activity-dependent *Bdnf* promoter IV were specifically reduced in cohesin-deficient cortical neurons after 6 hr activation with KCl (log 2 FC = –1.26 adj p=9.24e-05) as well as baseline conditions (log 2 FC = –1.39 adj p=1.19e-05, ***Figure 8b***). *Bdnf* promoter IV is located in the immediate vicinity of a strong CTCF peak in cortical neurons (***Bonev et al., 2017***; ***Figure 8c***, bottom track). This CTCF peak forms the base of a constitutive loop between *Bdnf* promoter IV and an activity-induced enhancer located ~2 Mb upstream of the *Bdnf* gene (***Figure 8c***). The strength of this loop was substantially reduced in cohesin-deficient neurons (***Figure 8c***, quantification in ***Figure 8d***). Hence, while the CTCF-based looping of *Bdnf* promoter IV to

a distant inducible enhancer is cohesin-dependent, the activity-regulated expression of the IEGs *Fos* and *Arc* is linked to enhancer-promoter loops that span limited genomic distances ( <40 kb) and can form independently of cohesin.

The formation of full-strength inducible loops at the *Bdnf* locus requires extended stimulation (6 hr, *Beagan et al., 2020*), which was not performed here. Formation of an inducible *Bdnf* loop was nevertheless discernible 1 hr after KCl stimulation (*Figure 8c and e*). Quantification shows that the inducible *Bdnf* loop increases in strength in response to KCl in wild-type but not in cohesin-deficient neurons (*Figure 8c and e*), indicating that formation of this loop requires cohesin. With ~1.68 Mb, the inducible *Bdnf* loop spans a substantially larger genomic distance than the enhancer-promoter loops at the IEGs *Fos* and *Arc* (*Beagan et al., 2020*). Because longer loops can be more difficult to detect due to distance-dependent background signal, we also analyzed loop strength corrected for the distance-dependent background signal. This analysis confirmed the cohesin-dependence of longer loops (*Figure 8—figure supplement 1*). The finding that long-range inducible loop formation at the *Bdnf* locus is cohesin-dependent supports the model that cohesin is required for the formation of longer chromatin loops.

## Discussion

Given that mutations in cohesin and CTCF cause intellectual disability in humans (*Deardorff et al., 2018*; *Gregor et al., 2013*; *Rajarajan et al., 2016*), the extent to which deficits in cohesin function alter neuronal gene expression remains a critical underexplored question. To define the role of cohesin in immature post-mitotic neurons we use experimental deletion of the cohesin subunit *Rad21* during a precise developmental window of terminal neuronal differentiation in vivo. We find impaired neuronal maturation and extensive downregulation of genes related to synaptic transmission, connectivity, neuronal development, and signaling in *Rad21*[lox/lox] *Nex*[Cre] neurons. Such gene classes are central to neuronal identity, and their wide-spread downregulation is likely to contribute to the observed maturation defects of *Rad21*[lox/lox] *Nex*[Cre] neurons. Acute proteolysis of RAD21-TEV corroborated a prominent role for cohesin in the expression of genes that facilitate neuronal maturation, homeostasis, and activation.

We have recently shown that cohesin loss in macrophages results in severe disruption of anti-microbial gene expression programs in response to macrophage activation (*Cuartero et al., 2018*). By contrast, cohesin loss only moderately affected genes constitutively expressed in uninduced and induced macrophages, supporting a model where cohesin-mediated loop extrusion is more important for the establishment of new gene expression than the maintenance of existing programs (*Cuartero et al., 2018*). Here, we extend this model to post-mitotic neurons in the murine brain. The two major ARG classes, IEGs, and LRGs, were broadly downregulated in cohesin-deficient neurons at baseline. However, IEGs and a subset of LRGs remained fully inducible by KCl and BDNF stimulation in the absence of cohesin. IEG-encoded transcription factors such as the AP1 members encoded by *Fos, FosB,* and *JunB* facilitate the induction of LRGs (*Malik et al., 2014*). Our data show that IEG-encoded transcription factors remain fully inducible in cohesin-deficient neurons. The failure to induce a subset of LRGs to wild-type levels is therefore not explained by a lack of IEG-encoded factors. This is in marked contrast to inducible gene expression in cohesin-deficient macrophages. A substantial number of LRGs in macrophages rely on the expression of early-induced interferons (IFN), which act in an autocrine and paracrine manner to support LRG induction (*Glass and Natoli, 2016*). Expression of IFN-dependent LRGs can be partially rescued by provision of exogenous IFN to cohesin-deficient macrophages (*Cuartero et al., 2018*).

The reliance of ARGs on cohesin for full activity-induced expression was linked to the scale of chromatin interactions as quantified by the analysis of Hi-C loops. The subset of LRGs that exhibit defects in inducibility in the absence of cohesin is characterized by longer-range chromatin loops than either IEGs or LRGs that remain fully inducible in the absence of cohesin. In addition to defective chromatin architecture, deregulated expression of a particular gene may also arise from disturbances in upstream regulatory mechanisms, such as the activity of specific signaling pathways, or the expression of particular transcription factors. We are not aware of signaling pathways that would unambiguously distinguish the subset of neuronal LRGs that are fully induced in the absence of cohesin from the subset of neuronal LRGs that remain deregulated in the absence of cohesin. As discussed above,

TFs encoded by IEGs are required for the induction of LRGs. Of note, these IEGs are fully induced in cohesin-deficient neurons.

The relationship between loop length and cohesin-dependence of neuronal gene expression extends to constitutively expressed cell type-specific neuronal genes. Neuronal genes related to synaptic transmission and glutamate receptor signaling that were downregulated in cohesin-deficient neurons also engaged in significantly longer chromatin loops than genes in the same GO terms that were not deregulated.

A subset of looping interactions made by ARGs and neuron-specific genes involve distal enhancers, suggesting that one role for cohesin in the expression of these genes may be to facilitate enhancer-promoter contacts. Our data indicate that specific enhancer-promoter loops at the key neuronal IEGs *Fos* and *Arc* can occur independently of cohesin in primary neurons. *Fos* enhancer-promoter loops remained responsive to environmental signals. Of note, *Fos* and *Arc* enhancer-promoter loops that were robust to cohesin depletion are relatively short-range ( <40 kb). By contrast, an inducible 1.68 Mb enhancer-promoter loop at the *Bdnf* locus, a constitutive chromatin loop between *Bdnf* promoter IV and an activity-induced enhancer, and all CTCF-based loops examined were substantial weakened in the absence of cohesin. While longer loops are inherently weaker than shorter loops, the observed differences in loop strength were robust to correction for the distance-dependent background signal. This indicates that our 5 C approach reliably quantifies the strength of both long and short loops.

In earlier studies, we found that contacts between the *Lefty1* promoter and the +8 kb enhancer, and between *Klf4* and enhancers at +53 kb remained intact in acutely cohesin-depleted ES cells (**Lavagnolli et al., 2015**). Synthetic activation of a *Shh* enhancer ~100 kb upstream of the TSS supported transcriptional activation of cohesin, while activation of a + 850 kb enhancer did not (**Kane et al., 2021**). Analysis of engineered enhancer landscapes in K562 cells indicates graded distance effects: Enhancers at ≥100 kb and 47 kb were highly and moderately dependent on cohesin, respectively, while loss of cohesin actually increased target gene transcription for enhancer distances ≤11 kb (**Rinzema et al., 2021**). Finally, promoter capture Hi-C in cohesin-depleted HeLa cells indicates ranges of $10^4$–$10^5$ bp for retained and $10^5$–$10^6$ bp for lost interactions (**Thiecke et al., 2020**). While these studies suggest that cohesin-dependence of chromatin contacts relates to genomic distance, they fail to link this observation to physiologically relevant gene expression. The new data described here demonstrate scaling of cohesin-dependence with genomic distance in primary neurons, and, importantly, link this finding to critical genome functions, specifically the implementation of cell type-specific gene expression programs during neuronal maturation and activation.

An open question concerns the mechanisms of enhancer-promoter contacts in the absence of cohesin. Current models of 3D genome folding posit competition between two forces, cohesin-mediated loop extrusion and condensate-driven compartmentalization (**Rao et al., 2017**; **Nora et al., 2017**; **Schwarzer et al., 2017**; **Beagan and Phillips-Cremins, 2020**). RNAP2, Mediator, the transactivation domains of sequence-specific transcription factors, and the C-terminal domain of the chromatin reader BRD4 are thought to support the formation of molecular condensates enriched for components of the transcriptional machinery (**Sabari et al., 2018**; **Boija et al., 2018**; **Rowley and Corces, 2018**; **Hsieh et al., 2020**). The activation of inducible ARG enhancers involves dynamic H3K27ac (**Beagan et al., 2020**), recruitment of RNA polymerases, and active transcription (**Joo et al., 2016**). Our data show that H3K27ac and transcription remain inducible at *Fos* enhancers in *Rad21*^lox/lox *Nex*^Cre neurons, which could potentially contribute to loop formation. At the *Arc* locus, the persistence of an enhancer-promoter loop in unstimulated cohesin-deficient neurons would be consistent with a role for cohesin-mediated loop extrusion in chromatin state mixing, and the separation of enhancer contacts (**Rao et al., 2017**). Notably, enhancer-promoter contacts resist the inhibition of BET proteins (**Crump et al., 2021**), selective degradation of BRD4 (**Crump et al., 2021**), Mediator (**El Khattabi et al., 2019**; **Crump et al., 2021**), RNA Polymerase II (**Thiecke et al., 2020**), or inhibition of transcription (**El Khattabi et al., 2019**). These observations suggest that numerous components of the transcriptional machinery may redundantly support associations between active genes and regulatory elements (**Sabari et al., 2018**; **Boija et al., 2018**). Nevertheless, phase separation-like forces provide an attractive hypothesis for the manner by which cohesin-independent enhancer-promoter loops may form.

In summary, our data demonstrate that the extent to which the establishment of activity-dependent neuronal gene expression programs relies on cohesin-mediated loop extrusion depends at least in part on the genomic distances traversed by their long-range chromatin contacts.

# Materials and methods

**Key resources table**

| Reagent type (species) or resource | Designation | Source or reference | Identifiers | Additional information |
|---|---|---|---|---|
| Genetic reagent (*Mus musculus*) | Rad21^lox Rad21^tm1.1Mmk | DOI: 10.1038/nature10312 | MGI:5293824 | |
| Genetic reagent (*Mus musculus*) | Nex^Cre Neurod6^tm1(cre)Kan | DOI: 10.1002/dvg.20256 | MGI:2668659 | |
| Genetic reagent (*Mus musculus*) | Rpl22(HA)^lox (RiboTag) Rpl22^tm1.1Psam | DOI: 10.1073/pnas.0907143106 | MGI:4355967 | |
| Genetic reagent (*Mus musculus*) | Rad21^tev Rad21^tm1.1Kktk | DOI: 10.1101/gad.605910 | MGI:4840469 | |
| antibody | anti-RAD21 (rabbit polyclonal) | Abcam | Cat #. ab154769 | WB: (dilution 1:1000) IF: (dilution 1:500) |
| antibody | anti-LAMIN B (goat polyclonal) | Santa Cruz Biotechnology | Cat #. sc-6216 | WB: (dilution 1:10,000) |
| antibody | anti-rabbit IgG (H + L) Alexa Fluor 680 (goat polyclonal) | ThermoFisher Scientific | Cat #. A-21109 | WB: (dilution 1:10,000) |
| antibody | anti-goat IgG (H + L) Alexa Fluor 680 (donkey polyclonal) | ThermoFisher Scientific | Cat #. A-21084 | WB: (dilution 1:10,000) |
| antibody | anti-HA (rabbit polyclonal) | Sigma | Cat #. H6908 | polysome immunoprecipitation |
| antibody | anti-GFAP (rabbit polyclonal) | Wako | Cat #. Z0334 | IF: (dilution 1:500) |
| antibody | anti-MAP2 (chicken polyclonal) | Abcam | Cat #. ab611203 | IF: (dilution 1:5000) |
| antibody | anti-GAD67 (mouse monoclonal) | Millipore | Cat #. MAB5406 | IF: (dilution 1:500) |
| antibody | anti-HA (mouse monoclonal) | Covance | Cat #. MMS-101R | IF: (dilution 1:1000) |
| antibody | IBA1 (rabbit polyclonal) | Wako | Cat #. 019–19741 | IF: (dilution 1:250) |
| antibody | anti-TUBB3 (Tuj1, mouse monoclonal) | Biolegend | Cat #. 801,202 | IF: (dilution 1:500) |
| antibody | anti-gamma-H2AX (rabbit polyclonal) | Bethyl Laboratories | Cat #. A300-081A | IF: (dilution 1:3000) |
| antibody | anti-Cleaved Caspase-3 (Asp175) (rabbit polyclonal) | Cell signalling | Cat #. 9,661 | IF: (dilution 1:400) |
| antibody | anti-TBR1 (rabbit polyclonal) | Abcam | Cat #. ab31940 | IF: (dilution 1:1000) |
| antibody | anti-CTIP2 (25B6, rat monoclonal) | Abcam | Cat #. ab18465 | IF: (dilution 1:500) |
| antibody | anti-CUX-1 (rabbit polyclonal) | Santa Cruz Biotechnology | Cat #. sc-13024 | IF: (dilution 1:400) |
| antibody | anti–Phospho-Histone H3 S10 Alexa Fluor 647 conjugate (rabbit polyclonal) | Cell signalling | Cat #. 9,716 | IF: (dilution 1:50) |
| antibody | anti-rabbit IgG (H + L) Alexa Fluor 647 (goat polyclonal) | ThermoFisher Scientific | Cat #. A-21244 | IF: (dilution 1:500) |
| antibody | anti-Rabbit IgG (H + L) Alexa Fluor 568 (goat polyclonal) | ThermoFisher Scientific | Cat #. A-11011 | IF: (dilution 1:500) |
| antibody | goat anti-mouse IgG (H + L) Alexa Fluor 488 (goat polyclonal) | ThermoFisher Scientific | A-11001 | IF: (dilution 1:500) |
| antibody | anti-chicken IgY (H + L) Alexa Fluor 568 (goat polyclonal) | Abcam | ab175711 | IF: (dilution 1:500) |
| Software, algorithm | ImageJ software | (http://imagej.nih.gov/ij/) | | |
| Software, algorithm | GraphPad Prism software | (https://graphpad.com) | | |
| Software, algorithm | FilamentTracer, Imaris software, Bitplane AG | https://imaris.oxinst.com | | |
| Software, algorithm | GSEA Desktop v3.0 | https://www.gsea-msigdb.org/gsea/index.jsp | | |
| Software, algorithm | Leica Application Suite X (LAS X, v2.7) software | https://www.leica-microsystems.com/products/microscope-software/p/leica-las-x-ls/ | | |

*Continued on next page*

*Continued*

| Reagent type (species) or resource | Designation | Source or reference | Identifiers | Additional information |
|---|---|---|---|---|
| Software, algorithm | CellProfiler v2.2 | https://cellprofiler.org | | |
| Software, algorithm | Image Studio Software (v5.2) | Li-cor Image Studio https://www.licor.com/bio/image-studio/ | | |

## Mice

Mouse work was done under a UK Home Office project license and according to the Animals (Scientific Procedures) Act. Mice carrying the floxed *Rad21* allele (*Rad21*<sup>lox</sup>, *Seitan et al., 2011*), in combination with the Cre recombinase in the Nex locus (*Goebbels et al., 2006*) and where indicated *Rpl22(HA)*<sup>lox/lox</sup> RiboTag (*Sanz et al., 2009*) were on a mixed C57BL/129 background. For timed pregnancies the day of the vaginal plug was counted as day 0.5. Genotypes were determined by PCR as previously reported (*Seitan et al., 2011*; *Sanz et al., 2009*). *Rad21*<sup>tev/tev</sup> mice have been described (*Tachibana-Konwalski et al., 2010*; *Weiss et al., 2021*).

## Neuronal cultures

For *Nex*<sup>Cre</sup> experiments, mouse cortices were dissected and dissociated from individual E17.5–E18.5 mouse embryos as described (*Beaudoin et al., 2012*) with minor modifications. Dissociated neurons were maintained in Neurobasal medium with B27 supplement (Invitrogen), 1 mM L-glutamine, and 100 U/mL penicillin/streptomycin for 10 days in vitro. Cells were plated at a density of $0.8 \times 10^6$ cells per well on six-well plates pre-coated overnight with 0.1 mg/ml poly-D-lysine (Millipore) and one third of the media in each well was replaced every 3 days. Cultures were treated with 5 µM Cytosine β-D-arabinofuranoside (Ara-C, Sigma) from day 2–4. For immunofluorescence staining neurons were plated on 12 mm coverslips (VWR) coated with poly-D-lysine at a density of $0.1 \times 10^6$ cells per coverslip. For cell-type-specific isolation of ribosome-associated mRNA, neurons from both cortices from each individual mouse embryo were seeded in a 10 cm dish.

For RAD21-TEV experiments, mouse cortices were dissected and dissociated on E14.5–15.5 as described (*Weiss et al., 2021*). Neurons were maintained in Neurobasal medium with B27 supplement (Invitrogen), 1 mM L-glutamine, and 100 U/ml penicillin/streptomycin. Cells were plated at a density of $1.25 \times 10^5$ /cm$^2$ on 0.1 mg/ml poly-D-lysine (Millipore) coated plates, and half the media was replaced every 3 days. Cultures were treated with 5 µM Ara-C at day 5. For cleavage of RAD21-TEV, neurons were plated as described above and transduced at day 3 with lentivirus containing ERt2-TEV at a multiplicity of infection of one. For ERt2-TEV dependent RAD21-TEV degradation, neurons were treated on culture day 10 with 500 nM 4-hydroxytamoxifen (4-OHT) or vehicle (ethanol) for 24 hr.

For KCl depolarization experiments, neuronal cultures were pre-treated with 1 µM tetrodotoxin (TTX, Tocris) and 100 µM D-(-)–2-Amino-5-phosphonopentanoic acid (D-AP5, Tocris) overnight to reduce endogenous neuronal activity prior to stimulation. Neurons were membrane depolarized with 55 mM extracellular KCl by addition of prewarmed depolarization buffer (170 mM KCl, 2 mM CaCl2, 1 mM MgCl2, 10 mM HEPES pH 7.5) to a proportion of 0.43 volumes per 1 ml volume of neuronal culture medium in the well. For BDNF induction experiments, neuronal cultures were treated with BDNF (50 ng/ml) for the indicated period of time at 10 days in vitro.

For Sholl analysis, dissociated cortical neurons were cultured as described (*Greig et al., 2013*). Astroglial monolayers were adhered to culture dishes and cortical neurons to coverslips, which were then suspended above the glia. Primary cultures of glial cells were prepared from newborn rat cortices. Four days before neuronal culture preparation, glial cells were seeded in 12-well plates at a density of $1 \times 10^4$ cells per well and one day before, the medium from the glial feeder cultures was removed and changed to neuronal maintenance medium for preconditioning. Mouse cortices were dissected from E17.5/E18.5 mouse embryos and kept up to 24 hr in 2 mL of Hibernate-E Medium (ThermoFisher) containing B27 supplement (Invitrogen) and 1 mM L-glutamine in the dark at 4 °C. Embryos were genotyped and the cortices from the desired genotypes were used to prepare neuronal cultures as described before. Neurons were plated on 24-well plates containing poly-D-lysine precoated 12 mm coverslips (VWR) at a density of $0.1 \times 10^6$ cells per well. Wax dots were applied to the coverslips, which served as 'feet' to suspend the coverslips above the glial feeder layer. 4 hr after neuronal seeding,

each coverslip containing the attached neurons was transferred upside down into a well of the 12-well dishes with the glial feeder. Cultures were treated with 5 µM Ara-C from day 2–4 and subsequently one third of the media was replaced every 3 days. For sparse neuronal GFP labeling, cortical neurons were transfected using 1 µg of peGFP-N1 plasmid along with 2 µl per well of Lipofectamine 2000 (Invitrogen) after 12 days in culture. Cultures were maintained for 14 days in vitro before fixation.

## RNA extraction and RT-qPCR

RNA was extracted with QIAshredder and RNeasy minikit (Qiagen). Residual DNA was eliminated using DNA-free kit (Ambion) and reverse-transcribed using the SuperScript first-strand synthesis system (Invitrogen). RT-PCR was performed on a CFX96 Real-Time System (Bio-Rad) with SYBR Green Master Mix (Bio-Rad) as per the manufacturer's protocol and normalized to *Ubc* and *Hprt* mRNA levels. Relative level of the target sequence against the reference sequences was calculated using the ΔΔ cycle threshold method. RT–PCR primer sequences:

| Gene | Forward (5'- 3') | Reverse (5'- 3') |
| --- | --- | --- |
| *Ubc* | AGGAGGCTGATGAAGGAGCTTGA | TGGTTTGAATGGATACTCTGCTGGA |
| *Hprt* | CCTGCTAATTTTACTGGCAACATCAACA | TTGAAATTCCAGACAAGTTTGTTGTTGG |
| *Rad21* | AGCACCAGCAACCTGAATGA | GATCGTCAAAGATGCCACCA |
| *Arc* | TACCGTTAGCCCCTATGCCATC | TGATATTGCTGAGCCTCAACTG |
| *Fos* | AATGGTGAAGACCGTGTCAGGA | TTGATCTGTCTCCGCTTGGAGTGT |
| *Cdkn1a* | GCAGACCAGCCTGACAGATT | GAGGGCTAAGGCCGAAGA |
| *Mdm2* | TGTGTGAGCTGAGGGAGATG | CACTTACGCCATCGTCAAGA |
| *Cdkn2a* | AATCTCCGCGAGGAAAGC | GTCTGCAGCGGACTCCAT |
| *Cdkn2b* | AGACTGCAAGCACGAAGAGG | TTGTCTTACTGGGTAGGGTTCAA |

## Protein analysis

Whole cell extracts were prepared by resuspending cells in PBS with complete proteinase inhibitor (Roche, Cat#18970600), centrifugation, and resuspension in protein sample buffer (50 mM Tris-HCl pH 6.8, 1% SDS, 10% glycerol) followed by quantification using Qubit. Following quantification 0.001% Bromophenol blue and 5% beta-mercaptoethanol were added. Sodium dodecyl sulphate-polyacrylamide gel electrophoresis (SDS-PAGE) was carried out with the Bio-Rad minigel system. 20 µg of protein sample and the benchmark pre-stained protein ladder (Biorad, #161–0374) were loaded on to a precast 10% polyacrylamide gel (Biorad, #456–1036). Resolved gels were blotted to a polyninylidene fluoride transfer membrane (Millipore, #IPVH00010) in transfer buffer (48 mM Trizma base, 39 mM glycine, 0.037% SDS, and 20% methanol) using the trans-blot semi-dry electro-phoretic transfer apparatus (BioRad). Membranes were incubated for 1 hr with fluorescent blocker (Millipore, HC-08) followed by primary antibody incubation diluted in blocker at an appropriate dilution for 2 hr or at room temperature or overnight at 4 °C. Primary antibodies were rabbit polyclonal to RAD21 (1:1000; ab154769, Abcam), goat polyclonal to LAMIN B (1:10,000; sc-6216; Santa Cruz Biotechnology), mouse monoclonal anti-myc tag (1:500, SC-40, Santa Cruz Biotechnology). Secondary antibodies were goat anti-rabbit IgG (H + L) Alexa Fluor 680 (1:10,000; A-21109, ThermoFisher), goat anti-mouse IgG, Alexa Fluor 680 1:10,000, and donkey anti-goat IgG (H + L) Alexa Fluor 680 (1:10,000; A-21084, ThermoFisher). Immobilon-FL PVDF membranes (Millipore) were imaged on an Odyssey instrument (LICOR).

## Cell-type-specific isolation of ribosome-associated mRNA

For polysome immunoprecipitation experiments, homogenates from 10 day cortical explant cultures were prepared as described (*Sanz et al., 2009*) with minor modifications. Cells were first washed two times on ice with 10 ml of PBS containing 100 µg/mL cycloheximide (Sigma). Cells were lysed in 50 mM Tris pH 7.5, 100 mM KCl, 12 mM MgCl2 (ThermoFisher), 1% IGEPAL CA-630 (Sigma), 1 mM DTT (Sigma), 200 U/mL RNasin (ThermoFisher), 1 mg/mL heparin (Sigma), 100 µg/mL cycloheximide, 1 x Protease inhibitor (Sigma) and homogenization with a motor-driven grinder and pestle for about

2 min. Samples were then centrifuged at 10,000 g for 10 min to create a postmitochondrial supernatant. For immunoprecipitations, 100 µL of Dynabeads Protein G (Invitrogen) were coupled directly to 10 µL of rabbit anti-HA antibody (Sigma, H6908). After polysome immunoprecipitation, total RNA was prepared using a RNeasy Plus Mini kit (Qiagen).

## RNAseq analysis

Total RNA was obtained in parallel from 10 day explant cultures of dissociated cortical neurons without stimulation (baseline); after overnight treatment with TTX and D-AP5 (TTX); and after overnight treatment with TTX and D-AP5 and depolarization with KCl for 1 hr (KCl1h) or 6 hr (KCl6h). RNA was extracted with QIAshredder and RNeasy mini kit (Qiagen). RNA-seq libraries were prepared from 600 ng of total RNA (RNA integrity number (RIN) >8.0) with TruSeq Stranded Total RNA Human/ Mouse/Rat kit (Illumina). For polysome immunoprecipitation experiments, 300 ng of total RNA was used for library preparation (RIN >9.0). RNA from Rad21-TEV neurons was purified with a PicoPure RNA Isolation kit (Applied Biosystems KIT0204), and 200 ng of total RNA was used to prepare libraries using the NEBNext Ultra II Directional RNA Library Prep Kit for Illumina (polyA enrichment), following the manufacturer recommendations. Library quality and quantity were assessed on a Bioanalayser and Qubit respectively. Libraries were sequenced on an Illumina Hiseq2500 (v4 chemistry) and at least 40 million paired end 100 bp reads per sample were generated per library and mapped against the mouse (mm9) genome. The quality of RNA-seq reads was checked by Fastqc (https://www.bioinformatics.babraham.ac.uk/projects/fastqc/) and aligned to mouse genome mm9 using Tophat version 2.0.11 (*Kim et al., 2013*) with parameters 'library-type = fr-first-strand'. Gene coordinates from Ensembl version 67 were used as gene model for alignment. Quality metrics for the RNA-Seq alignment were computed using Picard tools verion 1.90 (https://broadinstitute.github.io/picard/) (*Picard Toolkit, 2018*). Genome wide coverage for each sample was generated using bedtools genomeCoverageBed and converted to bigwig files using bedGraphToBigWig application from UCSC Genome Browser. Bigwig files were visualised using IGV. After alignment, number of reads on the genes were summarized using HTSeq-count (version 0.5.4; *Anders et al., 2015*). All downstream analysis was carried out in R (version 3.4.0). Differentially expressed genes between condition were determined using DESEq2 (*Love et al., 2014*). p-values calculated by DESeq2 were subjected to multiple testing correction using Benjamini-Hochberg method. Adjusted p-value of 0.05 was used to select the differentially expressed genes. Principal Component Analysis (PCA) and hierarchical clustering of samples were done on the normalized read counts (rlog) computed using DESeq2. KCl-inducible genes were defined as genes in $Rad21^{+/+}$ neurons with adjusted $p<0.05$ and log2 fold change $\geq 1$ in KCl1h versus TTX or KCl6h versus TTX. As reference we used previously defined activity dependent genes (*Kim et al., 2010*). Constitutive genes were defined as expressed genes in wild-type neurons with adj. $p \geq 0.05$ in KCl1h versus TTX and KCl6h versus TTX.

## Definition of ARGs and ARG classes

Our initial analysis used inducible ARGs described (*Kim et al., 2010*). The number of ARGs with assigned p-values across RNA-seq conditions was n=298 in the RiboTag RNA-seq of *Rad21* NexCre neurons and n=305 in the RNA-seq analysis of *Rad21* NexCre neurons. For the definition of ARG classes we used previously curated gene sets (*Tyssowski et al., 2018*). We refer to rapidly induced, translation-independent ARGs as early-induced IEGs (called rIEGs in *Tyssowski et al., 2018*) and late-induced LRGs (LRGs are called translation-independent delayed PRGs and translation-dependent SRGs by *Tyssowski et al., 2018*; *Beagan and Phillips-Cremins, 2020*; *Beagan et al., 2020*). The number of IEGs is n=19 (*Tyssowski et al., 2018*, their Supplementary Table 5). Of these, n=18 had assigned p-values in all conditions of our RNA-seq analysis, as well as informative Hi-C data, and were included in our analysis. The number of fully annotated LRGs as defined by *Tyssowski et al., 2018* (their Supplementary Table 5) is n=149 (comprised of 113 delayed translation-independent PRGs and n=36 translation-dependent SRGs). Of these, the number of LRGs with assigned p-values across RNA-seq conditions were n=107 in the RNA-seq analysis of *Rad21* NexCre neurons, n=101 in the RNA-seq analysis of RAD21-TEV neurons, and n=99 in the RNA-seq analysis of transiently RAD21-depleted and subsequently reconstituted neurons. For inclusion in the comparison of chromatin loop length versus gene expression in cohesin-deficient neurons, ARGs had to meet the following criteria: (i) assigned p-values across RNA-seq conditions, (ii) downregulation or no deregulation across TTX and

KCl conditions (ARGs that were downregulated in either TTX or KCl but not in both were excluded), and (iii) informative Hi-C data had to be available for the genomic region of each gene. These conditions were met by n=18 IEGs, n=22 downregulated LRGs, and n=43 non-deregulated LRGs.

## GO term analysis

GO terms enriched among differentially expressed genes were identified using goseq R package (*Young et al., 2010*) using all expressed genes in each comparison as background. GSEA was performed as described (*Subramanian et al., 2005*) using GSEA Desktop v3.0 (http://www.broadin-stitute.org/gsea). 'Wald statistics' from DESeq2 differential expression analysis were used to rank the genes for GSEA. The gene set collections C2 (curated; KEGG 186 gene sets) and C5 (GO ontologies; 5,917 gene sets) were obtained from Molecular Signature Database (MSigDB version 6.1; Broad Institute, http://www.broadinstitute.org/gsea/msigdb). Neuron-specific genes were identified using Neutools (*Gao et al., 2018*).

## Chromatin immunoprecipitation

ChIP was performed as described with minor modifications. Briefly, cells were cross-linked for 10 min at room temperature with rotation using 1% formaldehyde in cross-linking buffer (0.1 M NaCl, 1 mM EDTA, 0.5 mM EGTA and 25 mM HEPES-KOH, pH 8.0). The reaction was quenched using 125 mM glycine for 5 min with rotation and the cells were washed three times using ice-cold PBS containing complete protease inhibitor cocktail tablets (Roche). Cells were resuspended in lysis buffer (1% SDS, 50 mM Tris-HCl pH 8.1, 10 mM EDTA pH 8) with EDTA-free protease inhibitor cocktail and incubated for 30 min on ice. Cell lysates were then sonicated 20 times at 4 °C (Bioruptor Plus, Diagenode, 30/30 cycles) and centrifuged at 14,000 rpm for 1 min at 4 °C to remove cellular debris. 10% of total volume was taken as input. Input samples were reverse crosslinked overnight at 65 °C, then incubated for 1 hr with 9 mM EDTA pH 8, 3.6 mM Tris-HCl pH 6.8 and 36 µg/mL proteinase K at 45 °C. Input chromatin purification was performed using phenol-chloroform at 4 °C. Total chromatin was pre-cleared for 1 hr at 4 °C with rotation using protein A sepharose beads (P9424, Merck). 3 µg of anti-histone H3 (Abcam, ab1791) and 5 µg of anti-H3K27Ac (Active Motif, 39133) were added overnight at 4 °C with rotation. 100 µL of protein A sepharose beads were added to each IP for at least 4 hr before being washed with low salt buffer (150 mM NaCl, 2 mM Tris pH 8.1, 0.1% SDS, 1% Triton X-100 and 2 mM EDTA pH 8), high salt buffer (500 mM NaCl, 20 mM Tris pH 8.1, 0.1% SDS, 1% Triton X-100 and 2 mM EDTA pH 8), LiCl salt buffer (0.25 mM LiCl, 10 mM Tris pH 8.1, 1% NP-40, 1% sodium deoxycholate and 1 mM EDTA pH 8) and Tris-EDTA buffer (10 mM Tris pH 8.1 and 1 mM EDTA pH 8). Chelex-100 (Bio-Rad, catalog number #1421253) was added to the samples, which were then boiled and incubated for 1 hr at 55 °C with 36 µg/mL proteinase K and boiled once again. Samples were centrifuged at 12,000 rpm for 1 min and the beads washed once with nuclease-free water. The following PCR primers were used: Control region chr11: 71.177.622–71.177.792: forward, 5′-CATTCCAGGGCAACTCCACT-3′, reverse, 5′-CAGGGGCTCCTGTACTACCT-3′; *Fos* enhancer 1 forward, 5′-TCCGGTAAGGGCATTGTAAG-3′, reverse, 5′-CAAAGCCAGACCCTCATGTT-3′; *Fos* enhancer forward, 5′-TGCAGCTCTGCTCCTACTGA-3′, reverse, 5′-GAGGAGCAAGACTCCCACAG-3′.

## 3C template generation

Neuronal cultures were fixed in 1% formaldehyde for 10 min (room temp) via the addition (1:10 vol/vol) of the following fixation solution: 50 mM Hepes-KOH (pH 7.5), 100 mM NaCl, 1 mM EDTA, 0.5 mM EGTA, 11% Formaldehyde. Fixation was quenched via the addition of 2.5 M glycine (1:20 vol/vol) and scraped into pellets. Each pellet was washed once with cold PBS, flash frozen, and stored at –80 °C. For each condition, in situ 3 C was performed on two replicates of 4–5 million cells as described (*Beagan et al., 2020*). Briefly, cells were thawed on ice and resuspended (gently) in 250 µL of lysis buffer (10 mM Tris-HCl pH 8.0, 10 mM NaCl, 0.2% Igepal CA630) with 50 µL protease inhibitors (Sigma P8340). Cell suspension was incubated on ice for 15 min and pelleted. Pelleted nuclei were washed once in lysis buffer (resuspension and spin), then resuspended and incubated in 50 µL of 0.5% SDS at 62 °C for 10 min. SDS was inactivated via the addition of 145 µL H$_2$O, 25 uL 10% Triton X-100, and incubation at 37 °C for 15 min. Subsequently, chromatin was digested overnight at 37 °C with the addition of 25 µL 10 X NEBuffer2 and 100 U (5 µL) of HindIII (NEB, R0104S), followed by 20 min incubation at 62 °C to inactivate the HindIII. Chromatin was re-ligated via the addition of 100 µL 10%

Triton X-100, 120 µL NEB T4 DNA Ligation buffer (NEB B0202S), 12 µL 10 mg/mL BSA, 718 µL H$_2$O, and 2000 U (5 µL) of T4 DNA Ligase (NEB M0202S) and incubation at 16 °C for 2 hr. Following ligation nuclei were pelleted, resuspended in 300 µL of 10 mM Tris-HCl (pH 8.0), 0.5 M NaCl, 1% SDS, plus 25 µL of 20 mg/mL proteinase K (NEB P8107), and incubated at 65 °C for 4 hr at which point an additional 25 µL of proteinase K was added and incubated overnight. 3 C templates were isolated next day via RNaseA treatment, phenol-chloroform extraction, ethanol precipitation, and Amicon filtration (Millipore MFC5030BKS). Template size distribution and quantity were assessed with a 0.8% agarose gel.

## 5C library preparation

5 C primers which allow for the query of genome folding at ultra-high resolution but on a reduced subset of the genome were designed according to the double-alternating design scheme (*Kim et al., 2018b*; *Beagan et al., 2020*) using My5C primer design (*Lajoie et al., 2009*; http://my5c.umassmed.edu/my5Cprimers/5C.php) with universal 'Emulsion' primer tails. Regions were designed to capture TAD structures immediately surrounding the genes of interest in published mouse cortex Hi-C data (*Bonev et al., 2017*). The following regions were analyzed in this paper: chr2 107601077–110913077, chr10 107002896–109474896, chr12 86201802–87697802, chr15 73376037–74580037, chr17 89772409–92148409, chrX 97543400–98835400. 5 C reactions were carried out as previously described (*Beagan et al., 2020*). 600 ng (~200,000 genome copies) of 3 C template for each replicate was mixed with 1 fmole of each 5 C primer and 0.9 ug of salmon sperm DNA in 1 x NEB4 buffer, denatured at 95 °C for 5 min, then incubated at 55 °C for 16 hr. Primers which had then annealed in adjacent positions were ligated through the addition of 10 U (20 µL) Taq ligase (NEB M0208L) and incubation at 55 °C for 1 hr then 75 °C for 10 min. Successfully ligated primer-primer pairs were amplified using primers designed to the universal tails (FOR = CCTCTC TATGGGCAGTCGGTGAT, REV = CTGCCCCGGGTTCCTCATTC TCT) across 30 PCR cycles using Phusion High-Fidelity Polymerase. Presence of a single PCR product at 100 bp was confirmed via agarose gel, then residual DNA <100 bp was removed through Ampu-reXP bead cleanup at a ratio of 2:1 beads: DNA (vol/vol). 100 ng of the resulting 5 C product was prepared for sequencing on the Illumina NextSeq 500 using the NEBNext Ultra DNA Library Prep Kit (NEB E7370) following the manufacturer's instructions with the following parameter selections: during size selection, 70 µL of AMPure beads was added at the first step and 25 at the second step; linkered fragments were amplified using eight PCR cycles. A single band at 220 bp in each final library was confirmed using an Agilent DNA 1000 Bioanalyzer chip, and library concentration was determined using the KAPA Illumina Library Quantification Kit (#KK4835). Finally, libraries were evenly pooled and sequenced on the Illumina NextSeq 500 using 37 bp paired-end reads to read depths of between 11 and 30 million reads per replicate.

## 5C interaction analysis

5 C analysis steps were performed as described (*Gilgenast and Phillips-Cremins, 2019*; *Beagan et al., 2020*; *Fernandez et al., 2020*). Briefly, paired-end reads were aligned to the 5 C primer pseudo-genome using Bowtie, allowing only reads with one unique alignment to pass filtering. Only reads for which one paired end mapped to a forward/left-forward primer and the other end mapped to a reverse/left-reverse primer were tallied as true counts. Primer-primer pairs with outlier count totals, resulting primarily from PCR bias, were identified as those with a count at least 8-fold higher (100-fold for the lower-quality Arc region) than the median count of the 5 × 5 subset of the counts matrix centered at the primer-primer pair in question; outlier counts were removed.

Primer-primer pair counts were then converted to fragment-fragment interaction counts by averaging the primer-primer counts that mapped to each fragment-fragment pair (max of 2 if both a forward/left-forward and a reverse/left-reverse primer were able to be designed to both fragments and were not trimmed during outlier removal). We then divided our 5 C regions into adjacent 4 kb bins and computed the relative interaction frequency of two bins (i, j) by summing the counts of all fragment-fragment interactions for which the coordinates of one of the constituent fragments overlapped (at least partially) a 12 kb window surrounding the center of the 4 kb i$^{th}$ bin and the other constituent fragment overlapped the 12 kb window surrounding the center of the j$^{th}$ bin. Binned count matrices were then matrix balanced using the ICE algorithm[69,71] and quantile normalized across all eight replicates (two per condition) within each experimental set (neuronal activation and cohesin-rescue

experimental datasets were quantile normalized separately) as previously described (*Beagan et al., 2020*), at which point we considered each entry (i, j) to represent the Observed Interaction Frequency of the 4 kb bins i and j. Finally, the background contact domain 'expected' signal was calculated using the donut background mode (*Su et al., 2017*) and used to normalize the relative interaction frequency data for the background interaction frequency present at each bin-bin pair. The resulting background-normalized interaction frequency ('observed over expected') counts were fit with a logistic distribution from which p-values were computed for each bin-bin pair and converted into 'Background-corrected Interaction Scores' (interaction score = $-10*\log_2$(p-value)), which have previously shown to be informatively comparable across replicates and conditions (*Beagan et al., 2020*).

## Identification of neuronal enhancers

H3K27Ac ChIP-Seq and corresponding input datasets (*Malik et al., 2014*) were from NCBI GEO using accession GSE60192. The sra files were converted to fastq using sratoolkit and aligned to mouse genome mm9 using bowtie version 0.12.8 with default parameters (*Langmead et al., 2009*). Sequencing reads aligned to multiple positions in the genome were discarded. Duplicate reads were identified using Picard tools v 1.90 and removed from the downstream analysis. H3K27Ac peaks were identified using macs2 with 'broad' parameters (*Zhang et al., 2008*). Gene coordinates were obtained from Ensembl using 'biomaRt' R package (*Durinck et al., 2009*) and enhancers were assigned to nearest genes using 'nearest' function from GenomicRanges R package (*Lawrence et al., 2013*). Enhancers at Hi-C loop anchors were defined as reported previously (*Beagan et al., 2020*) and are provided there in Table S11.

## Hi-C loop span analysis

Loops were called on mouse cortical neuron Hi-C data (*Bonev et al., 2017*) as described (*Beagan et al., 2020*, loop calls are provided as Table S16 in *Beagan et al., 2020*). Cortical neuron CTCF ChIP-seq (*Bonev et al., 2017*) reads were downloaded from GEO accessions GSM2533876 and GSM2533877, merged, and aligned to the mm9 genome using Bowtie v0.12.7. Reads with more than two possible alignments were removed (-m2 flag utilized). Peaks were identified using MACS2 version 2.1.1.20160309 with a p value cutoff parameter of $10^{-4}$. Loops were classified as 'CTCF loops' if a CTCF peak fell within at least one anchor of the loop. Loops were classified as enhancer-promoter loops if the TSS of a gene of interest fell within one loop anchor and the other anchor of the same loop contained an activity-induced enhancer (for ARGs) or a constitutive or activity-induced enhancer (for neuronal genes related to synaptic transmission and glutamate receptor signaling; Table S11 in *Beagan et al., 2020*). The span of loops with an anchor that contained the TSS of a gene of interest was quantified by calculating the genomic distance between the midpoint of the loop's two anchors.

## Immunocytochemistry

Neurons plated on coverslips were fixed with warmed to 37 °C PBS containing 4% paraformaldehyde and 4% sucrose for 10 min at room temperature. Neurons were then permeabilized with 0.3% Triton X-100 for 10 min and treated with blocking solution (10% normal goat serum, 0.1% Triton X-100 in PBS) for 1 hr. Primary and secondary antibodies were diluted in 0.1% Triton X-100, 2% normal goat serum in PBS. Appropriate primary antibodies were incubated with samples for 2 hr at room temperature or overnight at 4 °C. Primary antibodies used were specific to RAD21 (1:500; rabbit polyclonal ab154769, Abcam), MAP2 (1:5000; chicken polyclonal ab611203, Abcam), GAD67 (1:500; mouse monoclonal MAB5406, Millipore), HA (1:1000; mouse monoclonal MMS-101R, Covance), GFAP (1:500; rabbit polyclonal Z0334, Dako), or IBA1 (1:250; rabbit polyclonal 019–19741, Wako). Secondary antibodies were incubated for 1 hr at room temperature. Goat anti-rabbit IgG (H + L) Alexa Fluor 647 (A-21244, ThermoFisher), Goat anti-Rabbit IgG (H + L) Alexa Fluor 568 (A-11011, ThermoFisher), goat anti-mouse IgG (H + L) Alexa Fluor 488 (A-11001, ThermoFisher), goat anti-chicken IgY (H + L) Alexa Fluor 568 (ab175711, Abcam) conjugates were used at a 1:500 dilution. Cells were mounted in Vectashield medium containing DAPI (Vector Labs).

Embryonic brains were fixed for 4 hr in 4% paraformaldehyde in PBS, washed in PBS, transferred to 15% sucrose in PBS for cryopreservation, embedded in OCT and stored at −80 °C until use. Coronal sections of 10 μm were cut with a Leica cryostat and mounted on glass slides. The sections were washed two times for 10 min in PBS, blocked with 0.3% TritonX-100, 5% normal goat serum in PBS,

at room temperature and incubated overnight with the primary antibody solution. Sections were then washed three times for 10 min each in PBS and were incubated for 1 hr in the dark with the secondary antibody solution. The primary antibodies were specific to RAD21 (1:500; rabbit polyclonal ab154769, Abcam), anti-gamma-H2AX (1:3000; rabbit polyclonal A300-081A, Bethyl Laboratories), Cleaved Caspase-3 (Asp175) (1:400; rabbit polyclonal 9661, Cell signalling), TBR1 (1:1000; rabbit polyclonal ab31940, Abcam), CTIP2 (1:500; rat monoclonal [25B6] ab18465, Abcam), CUX-1 (1:400; rabbit polyclonal sc-13024, Santa Cruz), and anti phospho-histone H3 (Ser10) Alexa Fluor 647 conjugate (1:50; rabbit polyclonal 9716, Cell signalling). The secondary antibodies used are described in the previous section. Sections were mounted with Vectashield medium containing DAPI.

### Confocal image analysis and quantification

For quantification analysis of RAD21 negative neurons and inhibitory neurons (GAD67+) images (1024 × 1024 pixels) were acquired using TCS SP5 confocal microscope (Leica Microsystems), using a HCX PL APO CS 40 x/1.25 lens at zoom factor 1 (373 nm/pixel). Images were acquired with identical settings for laser power, detector gain, and amplifier offset, with pinhole diameters set for one airy unit. DAPI-identified nuclei that colocalized with the GAD67 signal, or without RAD21 signal were counted as inhibitory or RAD21 negative neurons, respectively; and were quantified using a processing pipeline developed in CellProfiler (version 2.2, Broad Institute, Harvard, Cambridge, MA, USA, https://cellprofiler.org/). For astrocytes (GFAP+) and microglia (IBA1+) quantification in dissociated cortical neuronal cultures, the entire coverslips were imaged using a IX70 Olympus microscope equipped with a 4 × 0.1 NA Plan-Neofluar lens (1.60 μm/pixel) and the number of astrocytes and microglia were counted. The analysis and quantification of different cell types were done for three different experiments, each experiment containing at least two different samples of each genotype.

For Sholl analysis of dissociated cortical neurons, samples were imaged using a TCS SP8 confocal microscope, a HC PL APO CS2 40 x/1.30 lens at zoom factor 0.75 with a resolution of 2048 × 2048 pixel (189 nm/pixel, 0.5 μm/stack). Images were acquired with identical settings for laser power, detector gain, and amplifier offset, with pinhole diameters set for one airy unit. Approximately 10 neurons were imaged per sample and at least two different samples per genotype in each experiment; each experiment was performed three times. GFP +neurons were traced using the FilamentTracer package in Imaris software (Bitplane AG). Statistical significance was assessed using a repeat measures ANOVA with a Bonferroni Post Test (Prism-GraphPad Software).

## Acknowledgements

We thank Drs G Little, M Clements, and S Parrinello (University College London) for advice and practical instruction, S Di Giovanni and J Merkenschlager (Rockefelller University) for comments on the manuscript, members of our labs for helpful discussions, L Game for sequencing, and J Elliott and B Patel for cell sorting. This work was supported by the Medical Research Council UK, The Wellcome Trust (Investigator Award 099276/Z/12/Z to MM), EMBO (ALTF 1047–2012 TO LC), HFSP (LT00427/2013 to LC), the NIH National Institute of Mental Health (1R01-MH120269; 1DP1OD031253; JEPC), the NIH National Institute of Neural Disorders and Stroke (1R01-NS114226; JEPC), and a 4D Nucleome Common Fund grant (1U01DK127405; JEPC).

## Additional information

### Funding

| Funder | Grant reference number | Author |
|---|---|---|
| Medical Research Council | | Matthias Merkenschlager |
| Wellcome Trust | 099276/Z/12/Z | Matthias Merkenschlager |
| European Molecular Biology Organization | ALTF 1047-2012 | Lesly Calderon |
| Human Frontier Science Program | LT00427/2013 | Lesly Calderon |

| Funder | Grant reference number | Author |
|--------|------------------------|--------|
| National Institutes of Health | 1R01-MH120269 | Jennifer E Phillips-Cremins |
| National Institutes of Health | 1DP1OD031253 | Jennifer E Phillips-Cremins |
| National Institutes of Health | 1R01-NS114226 | Jennifer E Phillips-Cremins |
| 4D Nucleome Common Fund | 1U01DK127405 | Jennifer E Phillips-Cremins |

The funders had no role in study design, data collection and interpretation, or the decision to submit the work for publication. For the purpose of Open Access, the authors have applied a CC BY public copyright license to any Author Accepted Manuscript version arising from this submission.

## Author contributions

Lesly Calderon, Felix D Weiss, Jonathan A Beagan, Conceptualization, Formal analysis, Investigation, Visualization, Writing - review and editing; Marta S Oliveira, Formal analysis, Investigation, Writing - review and editing; Radina Georgieva, Formal analysis, Visualization, Writing - review and editing; Yi-Fang Wang, Thomas S Carroll, Data curation, Formal analysis, Methodology, Supervision, Visualization, Writing - review and editing; Gopuraja Dharmalingam, Data curation, Formal analysis, Methodology, Visualization; Wanfeng Gong, Investigation; Kyoko Tossell, Investigation, Methodology, Visualization; Vincenzo de Paola, Conceptualization, Methodology; Chad Whilding, Methodology; Mark A Ungless, Conceptualization, Methodology, Supervision; Amanda G Fisher, Conceptualization, Writing - review and editing; Jennifer E Phillips-Cremins, Conceptualization, Formal analysis, Methodology, Supervision, Visualization, Writing - original draft, Writing - review and editing; Matthias Merkenschlager, Conceptualization, Supervision, Visualization, Writing - original draft, Writing - review and editing

## Author ORCIDs

Lesly Calderon http://orcid.org/0000-0001-5253-7369
Felix D Weiss http://orcid.org/0000-0003-0228-5081
Jonathan A Beagan http://orcid.org/0000-0002-7049-3467
Jennifer E Phillips-Cremins http://orcid.org/0000-0002-4702-0450
Matthias Merkenschlager http://orcid.org/0000-0003-2889-3288

## Ethics

Laboratory bred mice of the appropriate genotype were maintained under SPF conditions and 12h light/dark cycle. Embryos were used to derive cells and tissues. Ethical approval was granted by the Home Office, UK, and the Imperial College London Animal Welfare and Ethical Review Body (AWERB).

## Decision letter and Author response

Decision letter https://doi.org/10.7554/eLife.76539.sa1
Author response https://doi.org/10.7554/eLife.76539.sa2

---

# Additional files

## Supplementary files

• Supplementary file 1. Gene expression analysis NexCre Ribotag RNA-seq Rad21 lox/lox minus wild-type.

• Supplementary file 2. Gene expression analysis total RNA-seq NexCre Rad21 lox/lox minus wild-type baseline ('at rest').

• Supplementary file 3. Gene ontology analysis of NexCre Ribotag RNA-seq Rad21 lox/lox minus wild-type.

• Supplementary file 4. Gene expression analysis RAD21-TEV 24 h cleaved vs control at baseline.

• Supplementary file 5. Gene expression analysis total RNA-seq NexCre Rad21 lox/lox minus wild-type TTX, KCl 1 h, KCl 6 h.

• Supplementary file 6. Gene expression analysis RAD21-TEV 24 h cleaved vs control BDNF 30 min

and BDNF 120 min.

- Transparent reporting form

## Data availability

RNAseq and 5C data generated in this study have been deposited at Gene Expression Omnibus under accession number GSE172429.

The following dataset was generated:

| Author(s) | Year | Dataset title | Dataset URL | Database and Identifier |
|---|---|---|---|---|
| Merkenschlager M | 2022 | Cohesin-dependence of neuronal gene expression relates to chromatin loop length | http://www.ncbi.nlm.nih.gov/geo/query/acc.cgi?acc=GSE172429 | NCBI Gene Expression Omnibus, GSE172429 |

The following previously published datasets were used:

| Author(s) | Year | Dataset title | Dataset URL | Database and Identifier |
|---|---|---|---|---|
| Malik AN | 2014 | Genome-wide identification and characterization of functional neuronal activity-dependent enhancers | https://www.ncbi.nlm.nih.gov/geo/query/acc.cgi?acc=GSE60192 | NCBI Gene Expression Omnibus, GSE60192 |
| Bonev B | 2017 | In-vitro differentiated cortical neurons | https://www.ncbi.nlm.nih.gov/geo/query/acc.cgi?acc=GSM2533876 | NCBI Gene Expression Omnibus, GSM2533876 |
| Bonev B | 2017 | In-vitro differentiated cortical neurons | https://www.ncbi.nlm.nih.gov/geo/query/acc.cgi?acc=GSM2533877 | NCBI Gene Expression Omnibus, GSM2533877 |

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
