## [Editor Report]

Neurons use activity-responsive gene programs to shape cell-specific identity and respond appropriately to environmental stimuli. By combining elegant protein degradation and cell-specific knockout approaches with transcriptional profiling and chromatin structure analysis, this manuscript delineates the contributions of cohesin (a key protein responsible for genome structure and organization), in developmental and activity-dependent gene expression programs as well as chromatin reorganization. These results demonstrate that cohesin is required for the full expression of key genes required for the maturation and activation of cortical excitatory neurons, and reveal a tight correlation between cohesin effects and the genomic distance of higher-order chromatin loops.

---

## [Decision Letter]

**Decision letter after peer review:**

Thank you for submitting your article "Reliance of neuronal gene expression on cohesin scales with chromatin loop length" for consideration by *eLife*. Your article has been reviewed by 3 peer reviewers, one of whom is a member of our Board of Reviewing Editors, and the evaluation has been overseen by Kevin Struhl as the Senior Editor. The reviewers have opted to remain anonymous.

Essential revisions:

1) All reviewers agreed that there is somewhat of a disconnect between effects of Rad21 depletion on activity-dependent gene expression and activity-dependent loop formation. While the manuscript presents some solid evidence that Rad21 contributes to SRG expression, it was not clear that this was due to prevention of new loops formed after stimulation, as the authors only report effects at constitutive loops. The manuscript would benefit from a more systematic comparison of the effects of RAD21 on inducible vs. constitutive loops, similar to the analysis that was performed with inducible vs. constitutive genes.

2) The conceptual framing of the manuscript could be improved by focusing more on developmental aspects of Rad21 depletion in neurons, and less on activity-dependent aspects. Given that many of the effects reported here at SRGs alter both basal and inducible gene expression, and the strong constitutive role for Rad21 in maintenance of chromatin architecture in neurons, this reframing would be a better fit for the results presented here.

3) The manuscript should include more thorough discussion to place the present work in context with prior work, as outlined in the individual reviews provided below.

4) A revised manuscript should address the individual concerns raised by reviewers to specify which gene sets are used for different analyses, clarify the terms IEG and SRG as used in the manuscript, and provide further analysis on the contribution of loop length to Rad21 effects.

*Reviewer #1 (Recommendations for the authors):*

1. Figure 5 convincingly shows that genes that are downregulated in RAD21 depleted neurons have longer HiC loop length, for both activity regulated genes and constitutive neuronal genes. However, in order to conclude that RAD21 effects on chromatin looping "scales with the genomic distance traversed by their chromatin contacts", it would seem to be necessary to conduct a more detailed analysis showing that the effects of RAD21 depletion directly correlate with the genomic distance of HiC contacts. By binning HiC lengths and showing average effects of RAD21 depletion on mRNA by bin classes (e.g,. 0-50, 50-100, 100-500, 500+kb loop lengths), it may be possible to address this question. As it stands, we only know that RAD21 downregulated genes tend to have lengthier loops, not that RAD21 effects scale with chromatin loop lengths.

2. The incorporation of 5C to show the lack of effect for RAD21 depletion on Fos enhancer-promoter looping (Figure 6) is encouraging and confirms the overall model of cohesin-dependent gene expression. However, the observations at Bdnf (a defined SRG, Supp. Figure 11) resemble those at Fos – only constitutive loops are lost following RAD21 depletion, with variable effects on gene basal levels and inducibility. Were the authors able to specifically examine activity-dependent loops that are formed involving SRGs as a result of stimulation? It does not appear that 5C at a more delayed timepoint (e.g. after 6hr of KCl stimulation) was conducted. This would have provided an opportunity to examine the stimulus-dependent loops formed at the Bdnf locus, previously identified by Beagan, et al., (PMID: 32451484). Either way, the manuscript could be improved by addition of the Bdnf figure (Supp. Figure 11) as a main figure in the text.

3. Prior work demonstrates that AP1 transcription factors serve as a key regulator of stimulus-activated enhancers in cortical neurons (PMID: 25195102). Similarly, evidence from ATAC-seq experiments suggests that IEG enhancers are already fully accessible prior to stimulation (PMID: 32810208). In contrast, AP1 members that comprise part of the IEG program are thought to serve as pioneer transcription factors (in combination with cell-selective TFs) to open chromatin at SRG enhancers. This manuscript would benefit from a more systematic comparison of the effects of RAD21 on inducible vs. constitutive loops, similar to the analysis that was performed with inducible vs. constitutive genes.

4. Many constitutive neuronal genes are also long genes, and these may be more likely to be disrupted after RAD21 depletion and resulting changes in genome structure and organization. This manuscript should include a supplementary analysis to determine whether RAD21 effects scale with gene length itself, in addition to chromatin loop length.

*Reviewer #2 (Recommendations for the authors):*

I feel that a revised version of the paper should address my comments listed here:

There should be a discussion on whether cohesin dependence is scaling with loop length in other cell types as well. For example, the Cuartero et al., 2018 study by some of the same authors, using approaches similar to the ones presented here in immune cells, and may be good dataset to compare with their findings in neurons.

Based on Figure 4a, only a tiny fraction of transcribed genes are immediate early genes IEGs (N=18) and only N=107 genes are secondary response genes. To me, it is not entirely clear how the Authors arrived at these numbers. The number of activity regulated genes should be in the several hundreds, based on the literature (including the Kim et al., 2010 paper that the Authors cite). In any case, one wonders what type of conclusion can be drawn from such a small set of N-18 IEG genes.

The Authors , if I understood correctly, probed multiple genomic loci with 5C, but specific information on the sequences and loci probed is hard to find in the paper. From reading the paper, it is clear that c-Fos and BDNF and Arc gene loci (Figures S9-11) were probed with 5C. Please be more specific on the area of genome interrogated with 5C.

The sensitivity of chromosome conformation capture techniques to detect loops and contacts declines with increasing linear genome distance between the anchors. There should be a discussion and if necessary, a control experiment (I believe in silico may suffice) whether or not the reported cohesion- sensitivity of longer range chromatin loopings could be reflective of the overall lower baseline signal of longer range chromosomal loopings. If so, this would be a significant confound for the Authors conclusion of cohesion dependence scaling with loop length.

*Reviewer #3 (Recommendations for the authors):*

My questions about the study fall in two parts.

First, I am left wondering about the significance of the observations and the way they are presented. The relationship between Rad21 dependence and longer loop length is supported by the data, but I struggled to understand what that might mean. Is there anything else the authors can add about why this might be the case? Is this special in neurons and other postmitotic cells and if so what might that mean? More importantly, the authors include a paragraph in the discussion about how promoter-enhancer interactions might work independent of cohesin, which is interesting, and perhaps it is a major finding of the study that those enhancers do not require Rad21. If this is the most important finding then the fact that this study reaches a different conclusion that Yamada 2019 needs to be directly addressed. That study also knocked out Rad21 in postmitotic neurons (CGNs) but did report disruption of promoter-enhancer loops and activity-induced gene expression. These data may well be stronger than those, but the authors should directly address the differences in the findings.

Second, given that the authors find the expression of many ion channels and synaptic proteins impacted by the loss of Rad21, they need to consider that any changes they see in impaired induction of activity-regulated genes – such as the reduced activation of Bdnf IV in Suppl Figure 11 – may arise from impaired intracellular signaling rather than a specific effect of Rad21 on the architecture of the Bdnf gene. Controls for the activation of signaling cascades and transcription factors could be added to address this concern. The demonstration that much of the effect of Rad21 knockout on activity-regulated genes is their reduction at baseline is certainly consistent with this idea of reduced synaptic drive to these genes, which is ongoing in cultures. KCl addition may largely overcome those deficits by providing such a strong drive through LVGCCs.

A few other questions:

1) When the authors refer to "activity-regulated genes" for example in Figure 2C, do they include only genes induced by neuronal activity or also those repressed by neuronal activity? This distinction is important because the meaning of those genes begin "Up" or "Down" in the Rad21 cKO is very different depending on the sign of their response to neuronal stimulation. This should be explicitly stated.

2) The data in Figure 3a on the distribution of different cell types in the cortex of Rad21 cKO mice is entirely anecdotal as presented. If the data are to be included they should be quantified and analyzed with appropriate statistics.

3) The terms IEG and SRG for the ARG classes are used here in a way that is not consistent with the literature. The division between the terms as the authors use them here seems to be time (IEGs early and SRGs late) whereas in the past the terms had more to do with mechanism – IEGs did not require stimulus-induced protein synthesis before they could be turned on whereas SRGs did. (Basically cycloheximide does not block IEG induction but it does block SRG induction). The Tyssowski paper referenced by the authors uses the language PRG for primary response genes that do not require protein synthesis for their induction, and then divide them into early and delayed PRGs for the reflection of time. BDNF is a delayed PRG and following the stimuli used here it does not require protein synthesis for its induction. The authors may want to consider clarifying their use of their terms depending on whether time or mechanism of transcription is their main focus to match other literature.

4) There is a paper from Kim Nasmyth's lab that knocked out Rad21 in postmitotic neurons of the fly with relatively severe phenotypes. By contrast at least the morphological phenotypes in the images in Suppl Figure 4 seem quite mild. This might be an important point of discussion if it is the case that the consequences of Rad21 function are different by species?

5) It is a little surprising that all the data from the final paragraphs on activity-regulated genes are in the supplementary figures. This points to evidence that the authors do not consider these to be the most important of their findings, which is slightly out of balance with the attention paid to activity-dependent genes in the text. Perhaps a rewrite of the paper would make the significance of the findings more obvious.

---

## [Author Response]

Essential revisions:1) All reviewers agreed that there is somewhat of a disconnect between effects of Rad21 depletion on activity-dependent gene expression and activity-dependent loop formation. While the manuscript presents some solid evidence that Rad21 contributes to SRG expression, it was not clear that this was due to prevention of new loops formed after stimulation, as the authors only report effects at constitutive loops.

To address this concern, the revised manuscript includes a new analysis of inducible *Bdnf* loop formation. While formation of the full strength inducible *Bdnf* loop requires 6h stimulation (Beagan et al., 2020, not performed in the present manuscript), an inducible *Bdnf* loop can nevertheless be seen to form 1h after KCl stimulation in wild-type neurons. The revised manuscript presents an analysis of this inducible *Bdnf* loop (revised Figure 8,). Quantification shows that the inducible *Bdnf* loop forms in wild-type but not in cohesin-deficient neurons, indicating that the formation of this loop requires cohesin. With ~1.68Mb, the inducible *Bdnf* loop spans a substantially larger genomic distance than the enhancer-promoter loops at the IEGs *Fos* and *Arc* (Beagan et al., 2020).

We have also followed the advice to re-balance data presentation, and have moved data on inducible gene expression from Supplementary to the main figures. In particular, data on the rescue of inducible gene expression, and the 5C analysis of the *Bdnf* locus are now presented in the main figures of the revised manuscript.

The manuscript would benefit from a more systematic comparison of the effects of RAD21 on inducible vs. constitutive loops, similar to the analysis that was performed with inducible vs. constitutive genes.

The most compelling inducible loops known in primary neurons are at *Fo*s and *Arc*, and our data show that these loops form independently of cohesin. As described above, the revised manuscript now includes an analysis of inducible *Bdnf* loops (Revised Figure 8).

As suggested by Referee 2, "the sensitivity of chromosome conformation capture techniques to detect loops and contacts declines with increasing linear genome distance". We have therefore performed a comparison of loop strength at different genomic distances (New Supplementary Figure 10). New Figure 8 —figure supplement 1 shows the observed loop strength after correction for the distance-dependent background signal (as described by Beagan et al., 2020). This approach allows an assessment of loop strength across distances. The y-axes are now on the same scale, allowing an appreciation of cohesin effects on loop strength, despite the differences in distance. This analysis indicates that the claimed differences in cohesin-dependence exist, independent of differences in loop length (Referee 2 Figure 2b and new Figure 8 —figure supplement 1).

2) The conceptual framing of the manuscript could be improved by focusing more on developmental aspects of Rad21 depletion in neurons, and less on activity-dependent aspects. Given that many of the effects reported here at SRGs alter both basal and inducible gene expression, and the strong constitutive role for Rad21 in maintenance of chromatin architecture in neurons, this reframing would be a better fit for the results presented here.

We have addressed this concern in two ways.

First, to strengthen the developmental aspects of the manuscript, we have added a quantitative analysis or cortical layer organisation: "While the total numbers of TBR1^+^ and CTIP2^+^ neurons were comparable in wild-type and *Rad21*^lox/lox^ cortices, TBR1^+^ and CTIP2^+^ neurons were reduced in the subplate and increased in layers 6 and 7 *Rad21*^lox/lox^
*Nex*^Cre^ cortices" (Revised Figure 3a).

Second, as discussed in detail above, the revised manuscript includes a new analysis of inducible *Bdnf* loop formation.

Third, we have followed the advice to re-balance data presentation, and have moved data on inducible gene expression from Supplementary to the main figures. In particular, data on the rescue of inducible gene expression, and the 5C analysis of the *Bdnf* locus are now presented in the main figures of the revised manuscript.

3) The manuscript should include more thorough discussion to place the present work in context with prior work, as outlined in the individual reviews provided below.

We have addressed this point by discussing our work in the context of experiments that depleted neuronal cohesin in flies and mice, as prompted by referee 3. We also emphasise that our data show an essential role of neuronal cohesin in mice, as indicated by lethality during the first 2 weeks of postnatal life. We have also followed the advice by referee 1 to discuss the role of AP1 transcription factors encoded by IEGs in the inducible expression of LRGs.

4) A revised manuscript should address the individual concerns raised by reviewers to specify which gene sets are used for different analyses, clarify the terms IEG and SRG as used in the manuscript, and provide further analysis on the contribution of loop length to Rad21 effects.

We agree that a clear definition of gene sets will benefit the manuscript. We used inducible ARGs described by Kim et al., 2010 for our initial analysis. The number of ARGs with assigned P-values across RNA-seq conditions was n = 298 in the RiboTag RNA-seq of *Rad21* NexCre neurons and n = 305 in the RNA-seq analysis of *Rad21* NexCre neurons.

For the definition of ARG classes we used gene sets curated by Tyssowski et al., 2018. We refer to rapidly induced, translation-independent ARGs as IEGs (these genes are called rIEGs in Tyssowski et al., 2018, and IEGs in Beagan et al., 2020). To describe late-induced ARGs we had initially adopted the term 'SRG' from Beagan et al., 2020. However, in light of the referees' comments we refer to late-induced ARGs as 'LRGs' in the revised manuscript. LRGs are called translation-independent delayed PRGs and translation-dependent SRGs by Tyssowski et al., 2018.

The number of IEGs as defined by Tyssowski et al., 2018, Supplementary Table 5 is n = 19. Of these, n = 18 had assigned P-values in all conditions of our RNA-seq analysis and informative Hi-C data, and were included in our analysis.

The number of fully annotated LRGs as defined by Tyssowski et al., 2018, Supplementary Table 5 is n = 149 (comprised of 113 delayed translation-independent PRGs and n = 36 translation-dependent SRGs). Of these, the number of LRGs with assigned P-values across RNA-seq conditions were n = 107 in the RNA-seq analysis of *Rad21* NexCre neurons, n = 101 in the RNA-seq analysis of RAD21-TEV neurons, and n = 99 in the RNA-seq analysis of transiently RAD21-depleted and subsequently reconstituted neurons.

For inclusion in the comparison of chromatin loop length versus gene expression in cohesin-deficient neurons, ARGs had to meet the following criteria: (i) assigned P-values across RNA-seq conditions, (ii) downregulation or no deregulation across TTX and KCl conditions (ARGs that were downregulated in either TTX or KCl but not in both were excluded), and (iii) informative Hi-C data had to be available for the genomic region of each gene. These conditions were met by n = 18 IEGs, n = 22 downregulated LRGs, and n = 43 non-deregulated LRGs.

We have added this information to the methods section of the revised manuscript.

Reviewer #1 (Recommendations for the authors):1. Figure 5 convincingly shows that genes that are downregulated in RAD21 depleted neurons have longer HiC loop length, for both activity regulated genes and constitutive neuronal genes. However, in order to conclude that RAD21 effects on chromatin looping "scales with the genomic distance traversed by their chromatin contacts", it would seem to be necessary to conduct a more detailed analysis showing that the effects of RAD21 depletion directly correlate with the genomic distance of HiC contacts. By binning HiC lengths and showing average effects of RAD21 depletion on mRNA by bin classes (e.g,. 0-50, 50-100, 100-500, 500+kb loop lengths), it may be possible to address this question. As it stands, we only know that RAD21 downregulated genes tend to have lengthier loops, not that RAD21 effects scale with chromatin loop lengths.

We agree with the referee. We have changed the title to "Cohesin-dependence of neuronal gene expression relates to chromatin loop length" and have revised other relevant passages of the manuscript accordingly.

2. The incorporation of 5C to show the lack of effect for RAD21 depletion on Fos enhancer-promoter looping (Figure 6) is encouraging and confirms the overall model of cohesin-dependent gene expression. However, the observations at Bdnf (a defined SRG, Supp. Figure 11) resemble those at Fos – only constitutive loops are lost following RAD21 depletion, with variable effects on gene basal levels and inducibility. Were the authors able to specifically examine activity-dependent loops that are formed involving SRGs as a result of stimulation? It does not appear that 5C at a more delayed timepoint (e.g. after 6hr of KCl stimulation) was conducted. This would have provided an opportunity to examine the stimulus-dependent loops formed at the Bdnf locus, previously identified by Beagan, et al., (PMID: 32451484). Either way, the manuscript could be improved by addition of the Bdnf figure (Supp. Figure 11) as a main figure in the text.

The revised manuscript includes a new analysis of inducible *Bdnf* loop formation. While formation of the full strength inducible *Bdnf* loop requires 6h stimulation (Beagan et al., 2020, not performed in the present manuscript), an inducible *Bdnf* loop can nevertheless be seen to form 1h after KCl stimulation in wild-type neurons. The revised manuscript presents an analysis of this inducible *Bdnf* loop. Quantification shows that the inducible *Bdnf* loop forms in wild-type but not in cohesin-deficient neurons, indicating that the formation of this loop requires cohesin. With ~1.68Mb, the inducible *Bdnf* loop spans a substantially larger genomic distance than the enhancer-promoter loops at the IEGs *Fos* and *Arc* (Beagan et al., 2020).

The finding that inducible looping at the *Bdnf* locus is cohesin-dependent adds substantially to the strength of our manuscript by supporting the correlative evidence that cohesin-dependent genes are characterised by longer loops. Based on this – and also the referee's advice – we have moved the revised *Bdnf* figure (formerly Supp. Figure 11) to a main of the revised manuscript (now Figure 8).

3. Prior work demonstrates that AP1 transcription factors serve as a key regulator of stimulus-activated enhancers in cortical neurons (PMID: 25195102). Similarly, evidence from ATAC-seq experiments suggests that IEG enhancers are already fully accessible prior to stimulation (PMID: 32810208). In contrast, AP1 members that comprise part of the IEG program are thought to serve as pioneer transcription factors (in combination with cell-selective TFs) to open chromatin at SRG enhancers. This manuscript would benefit from a more systematic comparison of the effects of RAD21 on inducible vs. constitutive loops, similar to the analysis that was performed with inducible vs. constitutive genes.

We thank the referee for these comments. The revised manuscript discusses the role of IEG-encoded TFs that facilitate LRG induction, citing the role of AP1 (PMID: 25195102; Malik et al., 2014) and the accessibility of IEG enhancers prior to stimulation (Carullo et al., 2020).

"Typical IEG enhancers such as *Fos* and *Arc* enhancers are fully accessible prior to stimulation (Carullo et al., 2020), and show increase H3K27ac and eRNA transcription in response to neuronal activation (Malik et al., 2014; Kim et al., 2010; Beagan et al., 2020; Joo et al., 2016; Carullo et al., 2020)"

and "IEG-encoded transcription factors such as the AP1 members encoded by *Fos, FosB* and *JunB* facilitate the induction of LRGs (Malik et al., 2014). Our data show that IEG-encoded transcription factors remain fully inducible in cohesin-deficient neurons. The failure to induce a subset of LRGs to wild-type levels is therefore not explained by a lack of IEG-encoded factors. This is in marked contrast to inducible gene expression in cohesin-deficient macrophages. A substantial number of LRGs in macrophages rely on the expression of early-induced interferons (IFN), which act in an autocrine and paracrine manner to support LRG induction (Glass and Natoli, 2016). Expression of IFN-dependent LRGs can be partially rescued by provision of exogenous IFN to cohesin-deficient macrophages (Cuartero et al., 2018)".

The most compelling inducible loops known in primary neurons are at *Fo*s and *Arc*, and our data show that these loops form independently of cohesin. As described in response to the referee's previous point, the revised manuscript now includes an analysis of an inducible loop at the *Bdnf* locus (Revised Figure 8).

4. Many constitutive neuronal genes are also long genes, and these may be more likely to be disrupted after RAD21 depletion and resulting changes in genome structure and organization. This manuscript should include a supplementary analysis to determine whether RAD21 effects scale with gene length itself, in addition to chromatin loop length.

It is an interesting suggestion that long neuronal genes might be more likely to be disrupted by loss of RAD21, but our analysis does not support this idea. (Author response image 1).

**Author response image 1. sa2fig1:** No scaling between gene length and cohesin-dependence in *Rad21* NexCre neurons.

Reviewer #2 (Recommendations for the authors):I feel that a revised version of the paper should address my comments listed here:There should be a discussion on whether cohesin dependence is scaling with loop length in other cell types as well. For example, the Cuartero et al., 2018 study by some of the same authors, using approaches similar to the ones presented here in immune cells, and may be good dataset to compare with their findings in neurons.

The comparison between neurons and macrophages suggested by the referee is very interesting. However, Hi-C data that is currently available for macrophages (49) has lower resolution than the Hi-C data we used for loop calling in cortical neurons (Bonev et al., 2017). As a result, fewer loops are found in macrophages than in cortical neurons (n = 3,029 in macrophages vs n = 24,937 loops in cortical neurons). Consequently, only a fraction of genes expressed in macrophages can be assigned associated loops. This hampers the quantitative comparison of loop lengths between macrophages and neurons, and between cohesin-dependent and -independent genes in macrophages. The findings described below should therefore be regarded as preliminary until this analysis can be repeated once higher-resolution macrophage Hi-C data becomes available.

– Based on the available data we find no difference in the length of loops associated with early- and late-inducible genes in macrophages. If confirmed by higher-resolution data, this would be consistent with the deregulation of both early- and late-inducible genes observed in resting and LPS-activated macrophages (Cuartero et al., 2018)

– Based on the available data we find chromatin loops associated with a significantly higher fraction of cohesin-dependent than with cohesin-independent genes in macrophages (P < 0.0001, odds ratio = 2.70). If confirmed by higher-resolution data, this would be consistent with a requirement for cohesin in the expression of genes with a high degree of 3D connectivity.

– Due to the resolution-related limitations of the macrophage Hi-C data, we believe that it would be premature to include these results in the revised manuscript.

Based on Figure 4a, only a tiny fraction of transcribed genes are immediate early genes IEGs (N=18) and only N=107 genes are secondary response genes. To me, it is not entirely clear how the Authors arrived at these numbers. The number of activity regulated genes should be in the several hundreds, based on the literature (including the Kim et al., 2010 paper that the Authors cite). In any case, one wonders what type of conclusion can be drawn from such a small set of N-18 IEG genes.

We agree that a clear definition of gene sets is required.

We used inducible ARGs described by Kim et al., 2010 for our initial analysis. The number of ARGs with assigned P-values across RNA-seq conditions was n = 298 in the RiboTag RNA-seq of *Rad21* NexCre neurons and n = 305 in the RNA-seq analysis of *Rad21* NexCre neurons.

For the definition of ARG classes we used gene sets curated by Tyssowski et al., 2018. We refer to rapidly induced, translation-independent ARGs as IEGs (these genes are called rIEGs in Tyssowski et al., 2018, and IEGs in Beagan et al., 2020). To describe late-induced ARGs we had initially adopted the term 'SRG' from Beagan et al., 2020. However, in light of the referees' comments we refer to late-induced ARGs as 'LRGs' in the revised manuscript. LRGs are called translation-independent delayed PRGs and translation-dependent SRGs by Tyssowski et al., 2018.

The number of IEGs as defined by Tyssowski et al., 2018, Supplementary Table 5 is n = 19. Of these, n = 18 had assigned P-values in all conditions of our RNA-seq analysis and informative Hi-C data, and were included in our analysis.

The number of fully annotated LRGs as defined by Tyssowski et al., 2018, Supplementary Table 5 is n = 149 (comprised of 113 delayed translation-independent PRGs and n = 36 translation-dependent SRGs). Of these, the number of LRGs with assigned P-values across RNA-seq conditions were n = 107 in the RNA-seq analysis of *Rad21* NexCre neurons, n = 101 in the RNA-seq analysis of RAD21-TEV neurons, and n = 99 in the RNA-seq analysis of transiently RAD21-depleted and subsequently reconstituted neurons.

For inclusion in the comparison of chromatin loop length versus gene expression in cohesin-deficient neurons, ARGs had to meet the following criteria: (i) assigned P-values across RNA-seq conditions, (ii) downregulation or no deregulation across TTX and KCl conditions (ARGs that were downregulated in either TTX or KCl but not in both were excluded), and (iii) informative Hi-C data had to be available for the genomic region of each gene. These conditions were met by n = 18 IEGs, n = 22 downregulated LRGs, and n = 43 non-deregulated LRGs.

We have added this information to the methods section of the revised manuscript.

The Authors , if I understood correctly, probed multiple genomic loci with 5C, but specific information on the sequences and loci probed is hard to find in the paper. From reading the paper, it is clear that c-Fos and BDNF and Arc gene loci (Figures S9-11) were probed with 5C. Please be more specific on the area of genome interrogated with 5C.

We apologise if the manuscript was unclear about the regions interrogated with 5C. We examined 6 different genomic regions:

chr Start End

2 107601077 110913077

10 107002896 109474896

12 86201802 87697802

15 73376037 74580037

17 89772409 92148409

X 97543400 98835400

These coordinates have been added to the Methods section:

"The following regions were analysed in this paper: chr2 107601077-110913077, chr10 107002896-109474896, chr12 86201802-87697802, chr15 73376037-74580037, chr17 89772409-92148409, chrX 97543400-98835400".

The revised figure legends now state the genomic coordinates of the regions examined.

The sensitivity of chromosome conformation capture techniques to detect loops and contacts declines with increasing linear genome distance between the anchors. There should be a discussion and if necessary, a control experiment (I believe in silico may suffice) whether or not the reported cohesion- sensitivity of longer range chromatin loopings could be reflective of the overall lower baseline signal of longer range chromosomal loopings. If so, this would be a significant confound for the Authors conclusion of cohesion dependence scaling with loop length.

We agree with the referee that "the sensitivity of chromosome conformation capture techniques to detect loops and contacts declines with increasing linear genome distance", and that it is important to ask whether or not the apparent cohesin-sensitivity of longer chromatin loops could arise simply because longer loops are weaker. We find that the observed strength of longer loops is consistently different between wild-type and cohesin-deficient neurons (New Figure 8 —figure supplement 1a, loops from Figure 1e), and between resting and activated wild-type neurons for the 1.68Mb inducible loop at the *Bdnf* locus (New Figure 8 —figure supplement 1a, loops from Figure 8e). At the *Fos* locus we observe a 155kb CTCF-based, cohesin-dependent loop, and two enhancer-promoter loops that are shorter (20-35kb) and remain fully inducible in cohesin-deficient cells. Despite differences in loop length, cohesin-dependent and cohesin-independent loops at the *Fos* locus are comparable in strength: The observed strength of the CTCF-based, cohesin-dependent loop in wild-type cells is ~1400. Both cohesin-independent inducible loops are comparable in strength to the CTCF-based, cohesin-dependent loop (~1650 in KCl^-^stimulated wild-type cells for the 35 kb loop between *Fos* and *Fos* enhancer 1, and ~2900 for the cohesin-independent inducible 20kb loop between *Fos* and *Fos* enhancer 2 in KCl^-^stimulated wild-type cells, New Figure 8 —figure supplement 1a, loops from Figure 6e).

Nevertheless, as stated by the referee, the observed values are highly biased by the distance-dependent background signal, the y-axes are on different scales, and the observed values for loop strength therefore need to be used with caution for claims regarding loop strength. New Figure 8 —figure supplement 1b therefore shows the observed loop strength after correction for the distance-dependent background signal (as described by Beagan et al., 2020). This approach allows an assessment of loop strength across distances. The y-axes are now on the same scale, allowing an appreciation of cohesin effects on loop strength, despite the differences in distance. This analysis indicates that the claimed differences in cohesin-dependence exist, independent of differences in loop length (New Figure 8 —figure supplement 1b). We have revised the discussion to state:

"While longer loops are inherently weaker than shorter loops, the observed differences in loop strength were robust to correction for the distance-dependent background signal. This indicates that our 5C approach reliably quantifies the strength of both long and short loops".

Of note, in addition to the cohesin sensitivity of loop formation itself, our study also links the degree to which gene expression is sensitive to cohesin with the length of chromatin loops that are formed by these genes. For this analysis, we measure deregulated gene expression by RNA-seq of wild-type and cohesin-deficient cells, and relate the results to the length of chromatin loops called in Hi-C data from wild-type cells. In this case, the issue of loop length versus loop strength does not arise, because (i) loops are called in wild type Hi-C and (ii) RNA-seq is not sensitive to loop length.

Reviewer #3 (Recommendations for the authors):My questions about the study fall in two parts.First, I am left wondering about the significance of the observations and the way they are presented. The relationship between Rad21 dependence and longer loop length is supported by the data, but I struggled to understand what that might mean. Is there anything else the authors can add about why this might be the case? Is this special in neurons and other postmitotic cells and if so what might that mean? More importantly, the authors include a paragraph in the discussion about how promoter-enhancer interactions might work independent of cohesin, which is interesting, and perhaps it is a major finding of the study that those enhancers do not require Rad21. If this is the most important finding then the fact that this study reaches a different conclusion that Yamada 2019 needs to be directly addressed. That study also knocked out Rad21 in postmitotic neurons (CGNs) but did report disruption of promoter-enhancer loops and activity-induced gene expression. These data may well be stronger than those, but the authors should directly address the differences in the findings.

Our findings indicate that short enhancer-promoter loops at the *Fos* and *Arc* loci form in the absence of cohesin, while a much longer 1.68Mb enhancer-promoter loop at the *Bdnf* locus requires cohesin (new data in revised Figure 8). Constitutive, CTCF-based loops also require cohesin. Correlative evidence shows that genes that are deregulated in cohesin-deficient neurons engage in longer loops than genes that are expressed at wild-type levels. This suggests – but does not prove – that the shorter loops associated with genes expressed at wild-type levels may remain intact in the absence of cohesin.

The ability of short enhancer-promoter loops to form – or at least persist – in the absence of cohesin does not appear to be unique to neurons or post-mitotic cells. Specifically, shorter enhancer-promoter loops are more likely to persist after acute cohesin depletion in HeLa cells than longer loops (Thiecke et al., 2020). Similarly, two recent preprints report short – but not long – enhancer-promoter loops in the absence of cohesin (Kane et al., 2021; Rinzema et al., 2021).

We suggest that our data do not contradict the findings reported by Yamada et al., 2019. The Yamada study lacked sufficient resolution for the analysis of individual loops, and the authors performed aggregate loop analysis instead. This approach would not have uncovered a subset of shorter loops that persist in the absence of cohesin. Similarly, the analysis of gene expression by Yamada et al., 2019 grouped genes into modules, and would not have uncovered the subset of inducible genes that remain inducible in the absence of cohesin.

Second, given that the authors find the expression of many ion channels and synaptic proteins impacted by the loss of Rad21, they need to consider that any changes they see in impaired induction of activity-regulated genes – such as the reduced activation of Bdnf IV in Suppl Figure 11 – may arise from impaired intracellular signaling rather than a specific effect of Rad21 on the architecture of the Bdnf gene. Controls for the activation of signaling cascades and transcription factors could be added to address this concern. The demonstration that much of the effect of Rad21 knockout on activity-regulated genes is their reduction at baseline is certainly consistent with this idea of reduced synaptic drive to these genes, which is ongoing in cultures. KCl addition may largely overcome those deficits by providing such a strong drive through LVGCCs.

We agree with the referee that defects in maturation and connectivity likely contribute to the deregulation of baseline gene expression in *Rad21* NexCre neurons, but can be partially are overcome by stimulation with KCl or BDNF. This is one of the reasons we performed parallel experiments with acutely cohesin-depleted neurons.

We are not aware of signaling pathways that would unambiguously distinguish the subset of neuronal LRGs that are fully induced in the absence of cohesin from the subset of neuronal LRGs that remain deregulated in the absence of cohesin. We have added to the discussion that TFs encoded by IEGs are required for the induction of LRGs, and that these IEGs are fully induced in cohesin-deficient neurons.

"IEG-encoded transcription factors such as the AP1 members encoded by *Fos, FosB* and *JunB* facilitate the induction of LRGs (Malik et al., 2014). Our data show that IEG-encoded transcription factors remain fully inducible in cohesin-deficient neurons. The failure to induce a subset of LRGs to wild-type levels is therefore not explained by a lack of IEG-encoded factors. This is in marked contrast to inducible gene expression in cohesin-deficient macrophages. A substantial number of LRGs in macrophages rely on the expression of early-induced interferons (IFN), which act in an autocrine and paracrine manner to support LRG induction (Glass and Natoli, 2016). Expression of IFN-dependent LRGs can be partially rescued by provision of exogenous IFN to cohesin-deficient macrophages (Cuartero et al., 2018)".

We agree that the failure of cohesin-deficient neurons to express certain genes at wild-type levels in may not necessarily result from defective chromatin architecture at each deregulated gene, but in some cases may arise from disturbances in upstream regulatory mechanisms, such as the activity of specific signaling pathways or the expression of particular transcription factors. We have added a Discussion section to acknowledge this:

"In addition to defective chromatin architecture, deregulated expression of a particular gene may also arise from disturbances in upstream regulatory mechanisms, such as the activity of specific signaling pathways, or the expression of particular transcription factors. We are not aware of signaling pathways that would unambiguously distinguish the subset of neuronal LRGs that are fully induced in the absence of cohesin from the subset of neuronal LRGs that remain deregulated in the absence of cohesin. As discussed above, TFs encoded by IEGs are required for the induction of LRGs. Of note, these IEGs are fully induced in cohesin-deficient neurons."

A few other questions:1) When the authors refer to "activity-regulated genes" for example in Figure 2C, do they include only genes induced by neuronal activity or also those repressed by neuronal activity? This distinction is important because the meaning of those genes begin "Up" or "Down" in the Rad21 cKO is very different depending on the sign of their response to neuronal stimulation. This should be explicitly stated.

We thank the referee for this important comment. We indeed focus our analysis on inducible activity-regulated genes. We now explicitly refer to "inducible ARGs" in the revised manuscript.

2) The data in Figure 3a on the distribution of different cell types in the cortex of Rad21 cKO mice is entirely anecdotal as presented. If the data are to be included they should be quantified and analyzed with appropriate statistics.

We agree with the referee. Figure 3a of the revised manuscript now presents a quantitative analysis of TBR1^+^ and CTIP2^+^ neurons in the subplate and in layers 5 and 6 (LV and VI) in control and *Rad21* NexCre brains at E16.5. We have added the following passage to the Results section:

"While the total numbers of TBR1^+^ and CTIP2^+^ neurons were comparable in wild-type and *Rad21*^lox/lox^ cortices, TBR1^+^ and CTIP2^+^ neurons were reduced in the subplate and increased in layers 6 and 7 *Rad21*^lox/lox^
*Nex*^Cre^ cortices."

3) The terms IEG and SRG for the ARG classes are used here in a way that is not consistent with the literature. The division between the terms as the authors use them here seems to be time (IEGs early and SRGs late) whereas in the past the terms had more to do with mechanism – IEGs did not require stimulus-induced protein synthesis before they could be turned on whereas SRGs did. (Basically cycloheximide does not block IEG induction but it does block SRG induction). The Tyssowski paper referenced by the authors uses the language PRG for primary response genes that do not require protein synthesis for their induction, and then divide them into early and delayed PRGs for the reflection of time. BDNF is a delayed PRG and following the stimuli used here it does not require protein synthesis for its induction. The authors may want to consider clarifying their use of their terms depending on whether time or mechanism of transcription is their main focus to match other literature.

We agree that a clear definition of gene sets will benefit the manuscript. We used inducible ARGs described by Kim et al., 2010 for our initial analysis. The number of ARGs with assigned P-values across RNA-seq conditions was n = 298 in the RiboTag RNA-seq of *Rad21* NexCre neurons and n = 305 in the RNA-seq analysis of *Rad21* NexCre neurons.

For the definition of ARG classes we used gene sets curated by Tyssowski et al., 2018. We refer to rapidly induced, translation-independent ARGs as IEGs (these genes are called rIEGs in Tyssowski et al., 2018, and IEGs in Beagan et al., 2020). To describe late-induced ARGs we had initially adopted the term 'SRG' from Beagan et al., 2020. However, in light of the referees' comments we refer to late-induced ARGs as 'LRGs' in the revised manuscript. LRGs are called translation-independent delayed PRGs and translation-dependent SRGs by Tyssowski et al., 2018.

The number of IEGs as defined by Tyssowski et al., 2018, Supplementary Table 5 is n = 19. Of these, n = 18 had assigned P-values in all conditions of our RNA-seq analysis and informative Hi-C data, and were included in our analysis.

The number of fully annotated LRGs as defined by Tyssowski et al., 2018, Supplementary Table 5 is n = 149 (comprised of 113 delayed translation-independent PRGs and n = 36 translation-dependent SRGs). Of these, the number of LRGs with assigned P-values across RNA-seq conditions were n = 107 in the RNA-seq analysis of *Rad21* NexCre neurons, n = 101 in the RNA-seq analysis of RAD21-TEV neurons, and n = 99 in the RNA-seq analysis of transiently RAD21-depleted and subsequently reconstituted neurons.

For inclusion in the comparison of chromatin loop length versus gene expression in cohesin-deficient neurons, ARGs had to meet the following criteria: (i) assigned P-values across RNA-seq conditions, (ii) downregulation or no deregulation across TTX and KCl conditions (ARGs that were downregulated in either TTX or KCl but not in both were excluded), and (iii) informative Hi-C data had to be available for the genomic region of each gene. These conditions were met by n = 18 IEGs, n = 22 downregulated LRGs, and n = 43 non-deregulated LRGs.

We have added this information to the methods section of the revised manuscript.

4) There is a paper from Kim Nasmyth's lab that knocked out Rad21 in postmitotic neurons of the fly with relatively severe phenotypes. By contrast at least the morphological phenotypes in the images in Suppl Figure 4 seem quite mild. This might be an important point of discussion if it is the case that the consequences of Rad21 function are different by species?

The observed changes may appear mild at first sight, but the data in Figure 3 —figure supplement 1a show that neuronal Rad21is required for postnatal viability in mice. We found *Rad21*^lox/lox^ NexCre mice at the expected Mendelian ratios up to the end of gestation, but none were alive by 2 weeks.

5) It is a little surprising that all the data from the final paragraphs on activity-regulated genes are in the supplementary figures. This points to evidence that the authors do not consider these to be the most important of their findings, which is slightly out of balance with the attention paid to activity-dependent genes in the text. Perhaps a rewrite of the paper would make the significance of the findings more obvious.

We thank the referee for this important comment. To re-balance the manuscript we have (i) strengthened the developmental aspects of the manuscript by adding the quantitative analysis or cortical layer organisation discussed above (ii) added new data on the formation of an inducible *Bdnf* loop (revised Figure 8) and (iii) moved important data into main figures, specifically *Bdnf* (revised Figure 8), and the rescue of inducible gene expression by cohesin (revised Figure 5d and 6b).